# The representation of abstract goals in working memory is supported by task-congruent neural geometry

**Mengya Zhang**, **Qing Yu***

Institute of Neuroscience, Key Laboratory of Brain Cognition and Brain-inspired Intelligence Technology, Center for Excellence in Brain Science and Intelligence Technology, Chinese Academy of Sciences, Shanghai, China

* qingyu@ion.ac.cn

## Abstract

Successful goal-directed behavior requires the maintenance and implementation of abstract task goals on concrete stimulus information in working memory. Previous working memory research has revealed distributed neural representations of task information across cortex. However, how the distributed task representations emerge and communicate with stimulus-specific information to implement flexible goal-directed computations is still unclear. Here, leveraging electroencephalography (EEG) and functional magnetic resonance imaging (fMRI) in human participants along with state space analyses, we provided converging evidence in support of a low-dimensional neural geometry of goal information congruent with a designed task space, which first emerged in frontal cortex during goal maintenance and then transferred to posterior cortex through frontomedial-to-posterior theta coherence for implementation on stimulus-specific representations. Importantly, the fidelity of the goal geometry was associated with memory performance. Collectively, our findings suggest that abstract goals in working memory are represented in an organized, task-congruent neural geometry for communications from frontal to posterior cortex to enable computations necessary for goal-directed behaviors.

## Introduction

In shifting environments, humans are capable of flexible cognition that relies on working memory, the ability to temporarily store and manipulate various kinds of information in mind [1]. These are not limited to basic sensory modalities but arguably more often involve abstract task information such as contingencies and contexts that guide goal-directed behaviors. When cooking without a recipe, one not only has to remember all the necessary ingredients (e.g., onions, celeries, and tomato, etc., for making a pasta), but the desirable state of each item (vegetables chopped, tomato crushed, and pasta cooked) and the action plans to achieve them (washing—chopping on a board—combining in a pan). All need to happen in a coordinated fashion. Yet within working memory research, the mechanisms of how neural codes of abstract task information and specific stimulus contents collectively support goal-directed behavior remains an open question.

**Data Availability Statement:** All data files are available from the Science Data Bank database (https://doi.org/10.57760/sciencedb.16868).

**Funding:** This work was supported by the Strategic Priority Research Program of the Chinese

Academy of Sciences (Grant No. XDB1010202), the Ministry of Science and Technology of China (STI2030-Major Projects 2021ZD0203701, 2021ZD0204202), the National Natural Science Foundation of China (32271089), CAS Project for Young Scientists in Basic Research (YSBR-071), Shanghai Pujiang Program (22PJ1414400) to Q.Y., and a Chinese International Postdoctoral Exchange Fellowship (YJ20220204) to M.Z.. The funders had no role in study design, data collection and analysis, decision to publish, or preparation of the manuscript.

**Competing interests:** The authors have declared that no competing interests exist.

**Abbreviations:** 2D, two-dimensional; CS, central sulcus; EEG, electroencephalography; EVC, early visual cortex; FA, flip angle; MRI, functional magnetic resonance imaging; FMT, frontomedial theta; GLM, general linear model; HRF, hemodynamic response function; IFG, inferior frontal gyrus; IFS, inferior frontal sulcus; iPCS, inferior precentral sulcus; ITG, inferior temporal gyrus; ITS, inferior temporal sulcus; LO, lateral occipital; LPFC, lateral prefrontal cortex; MFG, middle frontal gyrus; MTG, middle temporal gyrus; OFC, orbitofrontal cortex; OMPFC, orbital-medial prefrontal cortex; PCA, principal component analysis; PC, principal component; PCG, precentral gyrus; POS, parietal-occipital sulcus; RDM, representational dissimilarity matrix; ROI, region of interest; RSA, representational similarity analysis; STG, superior temporal gyrus; TE, time of echo; TR, time of repetition; wPLI, weighted phase lag index.

The coding schemes of low-level sensory information in working memory have been relatively well understood. Using multivariate decoding techniques or encoding models, stimulus-specific neural representations during memory delay has been found to be strongest in sensory cortices responsible for the initial processing of corresponding sensory features [2–9]. These results highlight sensory cortex as a crucial site for stimulus maintenance in working memory [10]. In parallel, mounting evidence has suggested a prominent role of frontal cortex in representing task information, with aggregated BOLD activity that reflects different levels of abstraction during cognitive control [11,12]. Neural activity in frontal cortex has also been found to preferentially encode higher-order, abstract representations of task contingencies [13], contexts [14], rules, or goals [15–17], providing top-down control signals that target stimulus representations in downstream brain regions [10,18]. Despite the general dichotomy, abstract task representations are not constrained to frontal cortex, but are also observed in posterior sensory cortex [16,17,19]. These distributed task representations are enhanced during goal implementation compared to pure goal maintenance, which are accompanied by increased long-range functional connectivity [20] and may reflect a spreading of task information from frontal cortex during the active execution of control processes. Nevertheless, how task representations emerge in distributed cortical areas and how they communicate with specific sensory information are still unclear.

Previous research has demonstrated how frontal and posterior cortex interact to implement task-based, top-down control. In particular, oscillatory coherence has been considered a potential neural mechanism for long-range information communication between brain areas [21,22]. For example, a body of work has demonstrated inter-areal, low-frequency oscillatory coherence as a mechanism for working memory maintenance of stimulus information [23–25]. It has remained less clear, however, whether task representations are also communicated in a similar vein. In parallel, it has been proposed that, to ensure stability in neuronal readouts and to enable task generalization and learning, population-level representations of task information may be compressed into a structured, low-dimensional neural subspace [26,27]. Different cortical areas can interact through a selective low-dimensional communication subspace, which may potentially serve as a population-level neural mechanism for information relay between cortical regions [28,29]. Moreover, there is also evidence that low-dimensional control representations can guide the flow of information across brain areas without directing high-dimensional, detailed information [30,31], possibly through low-frequency, theta-band oscillations for long-range communications [20,32]. The findings summarized above suggest a potential mechanism by which abstract task information is organized in a structured, low-dimensional representational format and relayed across cortical areas through subspace communication, possibly between frontal cortex and downstream sensory areas through oscillatory mechanisms to support goal implementation.

In the present study, we set out to directly test this hypothesis by tracking the emergence of abstract task representations in different cortical regions and investigating how they interacted with specific stimulus contents during human working memory. In particular, we focused on goal information, a well-established form of abstract task information. We developed a principled approach to construct goal and stimulus representations in a structured manner, by designing a goal and a stimulus space, each defined by 2 orthogonal dimensions that formed a theoretical space. This design allowed for a direct examination of the congruency between the neural representational geometries and the theoretical task structure, similar to how stimulus-specific representational subspaces were uncovered [33–36]. In 2 separate studies using electroencephalography (EEG) and functional magnetic resonance imaging (fMRI), healthy participants performed a novel delayed-recall task that required the maintenance of both a goal and a specific stimulus and at a later stage the manipulation of the stimulus features based

on the goal. Combining EEG with state space analyses, we found that task representations consistent with the task-congruent, low-dimensional goal space first appeared in frontal activity patterns before emerging posteriorly. Simultaneously the stimulus-related neural representations were present in posterior activity. The strength of the task-congruent goal geometry was associated with memory performance. Moreover, the transfer of goal representations from frontal to posterior regions and the interactions between goal and stimulus information were modulated by frontomedial theta-to-posterior coherence. Finally, using fMRI, the frontal task-congruent goal geometry was localized to subregions in lateral prefrontal (LPFC) and orbital-medial prefrontal cortex (OMPFC), and the posterior goal geometry, along with the corresponding stimulus geometry, was localized to posterior visual-related regions. Together, these 2 studies provided converging evidence that task-congruent representational geometry of goal information emerges in frontal cortex and transfers to posterior cortex for implementation, giving rise to successful goal-directed behavior.

## Results

### EEG behavioral results

Participants ($n$ = 22) performed a working memory manipulation task alongside with EEG recording. The task required participants to mentally maintain a memory goal and a visual stimulus and then manipulate the stimulus according to the specific goal. Memory stimuli varied along 2 orthogonal feature dimensions (Fig 1B, right), size (from small to big) and color (from green to red). Similarly, memory goals (Fig 1B, left) also varied along 2 orthogonal goal dimensions, size (adjusting smaller to adjusting bigger) and color (adjusting greener to adjusting redder) goals. In other words, memory stimuli formed a two-dimensional (2D) stimulus space, and memory goals formed a 2D goal space. Prior to the main task, participants first learned the degree of required adjustment for all possible stimuli in the working memory task, such that for any given stimulus in the 2D stimulus space, the correct degree of adjustment following a specific goal was pre-learned (Fig 1C). This design allowed for the dissociation between goal and stimulus information, which means that for different stimuli, the same goal would lead to different correct responses. On each trial, participants were first cued with one of 4 possible goals (Goal cue: bigger and redder, bigger and greener, smaller and redder, smaller and greener; Fig 1A) and maintained the goal over the first delay period (Delay 1). After that they were presented with a to-be-memorized stimulus of a specific size and color (Sample), and maintained both the goal and stimulus over a second delay for goal implementation (Delay 2), before starting to make adjustments during the response period (Response; Fig 1E). A delay period was included in both stages to better dissociate memory-related activity from sample- or response-driven activity.

Memory errors were calculated by subtracting correct answers from responses. Mean size error (Fig 1F) was −0.2% of starting size (SD = 4%) and was not significantly different from 0 ($t(21)$ = 0.22, $p$ = 0.82). Mean color error was 1.77 color steps (SD = 0.68) and was larger than 0 ($t(21)$ = 12.17, $p < 0.001$). These suggested participants were able to perform the task according to the instructions and memorize the stimulus attributes, as there was no bias in size response and a small bias in color response towards redness, compared to the entire available color range (120 steps). We also showed the individual distributions of errors (S1 Fig). Importantly, we used the absolute values of the response error in all subsequent behavioral correlation analyses, as the raw values were signed and only the magnitude itself measured performance. Mean absolute size error was 13.3% of starting size (SD = 2.6%), while mean absolute color error was 10.1 steps (SD = 1.81).

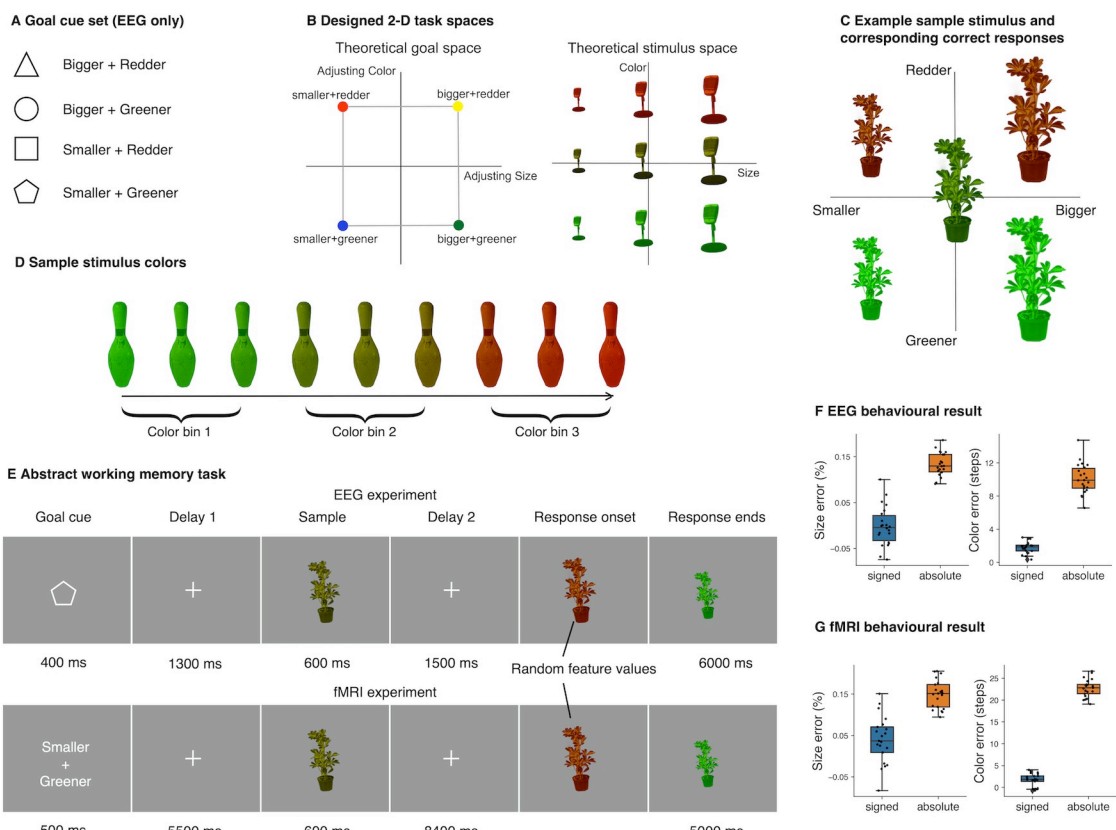

**Fig 1. Task schematics and behavioral results.** **(A)** Associations between shape cues and task goals. For EEG experiment, cues were used instead of text prompts for indicating task goals. Pairings did not change across participants. **(B)** Theoretical 2D goal and stimulus spaces. To directly examine the neural geometries of working memory representations, both goal and stimulus were constructed from 2 orthogonal axes. The task goal space consisted of adjusting size and color while the stimulus space consisted of continuous stimulus size and color (red-greenness). Stimulus size and color were grouped into three bins each for subsequent analyses. **(C)** Example sample stimulus (center) and the corresponding correct answers (4 quadrants) according to the task goals. The distances between any given sample and correct answers in terms of feature values were fixed and pre-learned by participants in a separate behavioral session. **(D)** Illustration for all starting values in sample stimulus color. **(E)** Schematics of EEG and fMRI paradigms. The abstract working memory task required the memorization of task goal and stimulus features, and at response phase, the manipulation of stimulus based on the cued goal. At the beginning of response, the appearance of a stimulus on screen marked the response onset; however, the size and color of this initial item were set with random values to prevent motor planning. **(F, G)** Size and color errors for EEG and fMRI experiments, respectively. The response errors were calculated by subtracting correct values from response. Mean signed error (blue) showed the degree of bias in participants' responses, whereas mean absolute error (orange) showed the degree of precision. Individual point represents mean error of each participant. Error bars denote 1.5 IQR. Data and code that support these findings are available at: https://doi.org/10.57760/sciencedb.16868. EEG, electroencephalography; fMRI, functional magnetic resonance imaging; IQR, inter quantile range.

## Task-congruent, two-dimensional goal representational geometry transfers from frontal to posterior channels

Having established behaviorally that participants could well maintain the remembered goals and stimuli, and manipulate the stimuli according to the remembered goal in the designed 2D spaces, we next sought to explore the structures of goal and stimulus representations in the neural state space and their alignment with the designed 2D task spaces. Recent work has successfully utilized dimension reduction techniques to reveal neural subspaces and representational geometries for stimulus information in working memory, including on human neuroimaging data [34–37]. This approach transcends the mere confirmation of whether information is encoded without specifying its representational format (e.g., via multivariate

decoding methods) and can provide richer descriptions of the coding principles adopted by neuronal populations in retaining and utilizing relevant information [38]. The experiment design featuring 2 orthogonal dimensions of task goals allows us to effectively characterize task-congruent, 2D representational geometry of goal information at the neural level. To this end, for each individual's neural data, we used principal component analysis (PCA) to identify the 2 principal components (PCs) that explained the most variance in terms of task goals. We then examined whether the 2D structural information in goals could be reflected in the 2D neural subspaces as revealed by the PCAs. To quantify the degree of which the resulting representational geometry matched the theoretical square-like 2D structure, we computed a measure called circularity index (*C*), defined as the ratio between a geometric shape's area and the square of the total length of perimeter. A larger circularity index would indicate higher similarity between the 2 structures. Time-resolved individual circularity indices (smoothed with a non-overlapping 80-ms sliding window) were calculated over the course of the trial. Significance of the time courses was evaluated using cluster-based permutation tests (details in Methods). For visualization purposes only, separate PCAs based on group-concatenated data were performed and projected coordinates corresponding to unique goals were connected in the same order as in the designed goal space.

To examine representations of goal and stimulus, we primarily focused our subsequent analyses on 2 sets of EEG channels, frontal and posterior channels (Fig 2A). The result showed that the hypothesized 2D goal geometry was present in both groups of channels albeit at different time periods over the course of a trial: frontal channel activities exhibited a neural geometric structure that followed the theoretical 2D task space (i.e., adjusting size and color) during cue presentation (Goal cue; 0–320 ms; Fig 2B, top panel) and goal maintenance (Delay 1; 960–1,440 ms). By contrast, in posterior channels (Fig 2B, bottom panel), activity patterns did not exhibit the representational structure predicted by the task space, until towards the end of goal implementation (Delay 2; 3,580–4,300 ms), extending to the response phase (Response). Visualization of group-level results confirmed the 2D goal geometry in corresponding task epochs in the neural subspaces defined by PCA (Fig 2C). These results naturally bear the question of whether the 2 neural geometries were related or independently formed. To provide insight into this, we included the rest of the channels that sit between frontal and posterior channels (i.e., central channels) in order to compare the temporal orders of the emergence of 2D goal geometries among these channel groups. We found that central channels exhibited significant 2D goal geometry as early as the middle of stimulus presentation (Sample; 2,140–4,060 ms; Fig 2B, middle panel), which followed the frontal geometry but preceded posterior activities. Therefore, this suggests the 2D goal geometry was first formed in frontal channels and emerged over time to posterior channels, possibly by traveling backwardly through central channels.

To ensure the validity of the individual circularity index, we compared it with outcomes derived from another more traditional metric, namely representational similarity analysis (RSA). Using the task-congruent 2D goal model (S2A Fig, left), only during Delay 1 in frontal channels, and late Delay 2 onwards for posterior channels, time-resolved RSA correlations were significant (S2B Fig), in line with the 2D goal geometry revealed by our geometry analysis. Thus, we confirmed that the results above were robust and commensurable with the alternative analytic approach of RSA. It should be noted that there are conceptual differences between these 2 approaches: while circularity index measures how similar the geometry captured in the first 2 PCs is to a 2D square-like structure, RSA measures the overall pairwise distances across all data dimensions. Moreover, given that the 4 available button-feature adjustment mappings remained the same across trials and participants, and that motor response and task goals were indeed related according to a post hoc analysis ($\chi^2$ = 588.2,

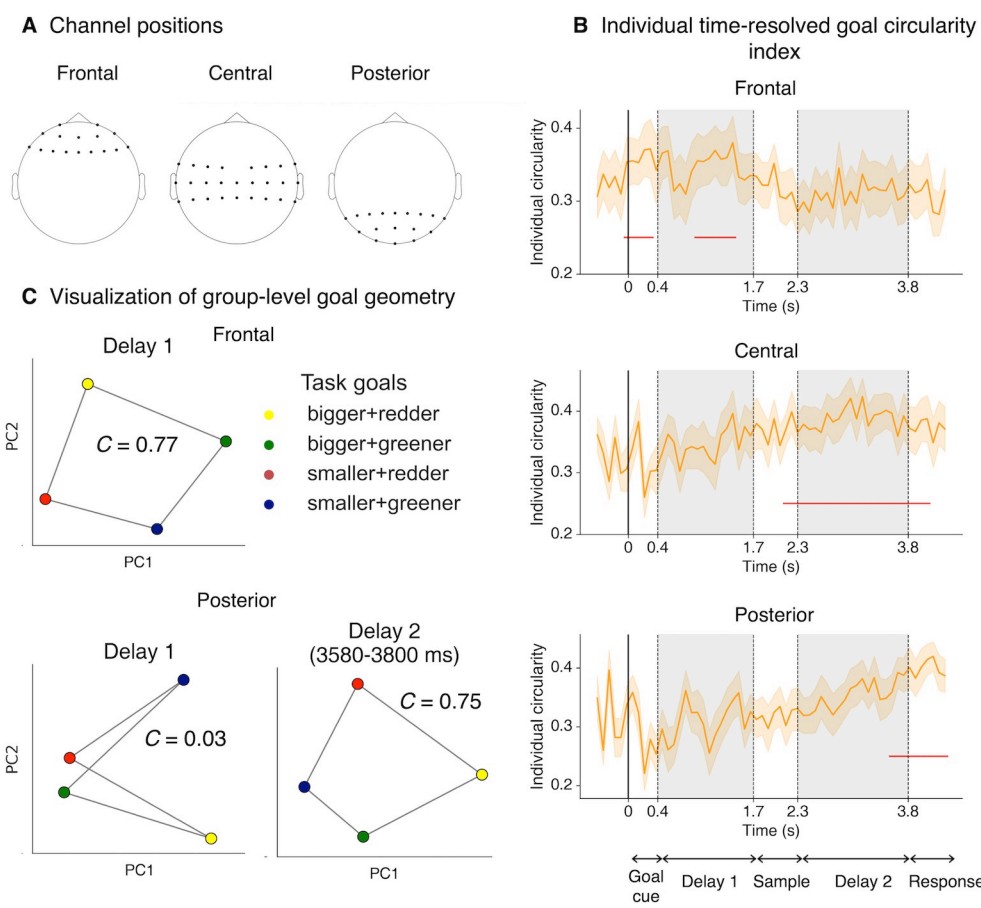

**Fig 2. Two-dimensional representational geometries of task goals. (A)** Demonstration of frontal, central, and posterior channel locations. **(B)** Time courses of individual circularity index in frontal (top), central (middle), and posterior (bottom) channels, using data averaged over a non-overlapping 80-ms sliding window. Error bars denote standard error of the mean (SEM) using the number of participants as the degree of freedom. Red horizontal lines denote time points when 2D goal geometry was significant using a cluster-based permutation test ($\alpha = 0.05$). **(C)** Group-level visualization of goal-specific neural space. Task goal conditions were projected onto the 2D neural subspace identified by group-level PCA from averaged Delay 1 epoch for frontal channels (top) and posterior channels for Delay 1 and late Delay 2 (bottom), guided by significant time points in the time course results. Coordinates were connected in the order according to the conceptual task space. *C* denotes the circularity index. Data and code that support these findings are available at: https://doi.org/10.57760/sciencedb.16868. 2D, two-dimensional; PCA, principal component analysis; SEM, standard error of the mean.

$p < 0.001$), to rule out the possibility that motor preparations caused the observed 2D representations, we calculated the time-resolved motor-specific circularity index as a control analysis. There was no time period during which frontal or posterior activities exhibited significant 2D motor geometry (S3A Fig).

To examine whether other representational formats of goal information coexisted, we specified an additional theoretical model, i.e., a conjunctive model which assumes task goals are equidistant from each other (S2A Fig, right), and calculated its similarities to neural data. In addition, we estimated both the 2D and conjunctive models jointly in a competitive manner to further decoupled the results due to their correlated nature (see Methods). It was observed that the conjunctive model was instead most prominent in posterior channels during Goal cue and Delay 1 (S2C and S2E Fig), likely related to sensory-driven signals from the visual cue.

## Strength of task-congruent goal geometry was associated with memory performance

Having established the existence of the 2D goal-specific representations in both frontal and posterior channel activities, we tested whether the strength of such task-congruent neural geometry was related to behavior. Firstly, correlational analyses were performed between individual circularity indices and memory performance as measured by the sum of standardized absolute color and size errors. We hypothesized that a stronger 2D goal geometry would lead to a better separation of conditions for downstream populations and more robust spatial transmission of information, hence better performance should be observed. Given that significant 2D geometry existed during Goal cue and part of Delay 1 in frontal channels and during part of Delay 2 and Response in posterior channels, electrophysiological activities were averaged within each epoch and channel groups respectively to compute individual circularity indices and correlated with behavioral performance. All subsequent correlation results were corrected using the FDR method, unless specified. 2D goal geometry in frontal channels showed a trending negative correlation with response error during Delay 1 (Spearman's correlation; $r =$ −0.40, $p = 0.064$; Fig 3A), but not in Goal cue ($r = 0.18$, $p = 0.79$). Conversely, we did not find any relationship for posterior activities during both Delay 2 ($r = 0.24$, $p = 0.86$) or Response ($r = 0.001$, $p = 0.86$).

Next, we performed within-subject comparisons which afford more sensitivity than between-subject methods, as they are less subject to individual differences beyond the scope of current considerations but still may affect task performance. For goal geometry, we took the

**A** Frontal 2-D goal geometry and behavior correlation (Delay 1)

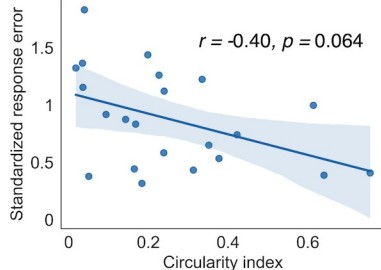

**B** Difference in frontal 2-D goal geometry between upper and lower quartile trials

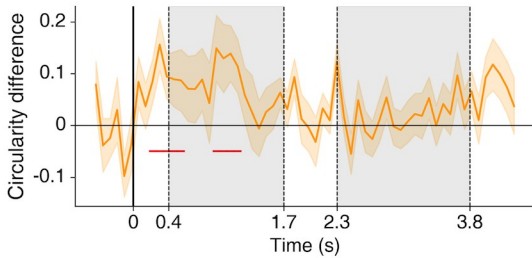

**C** Difference in posterior 2-D goal geometry between upper and lower quartile trials

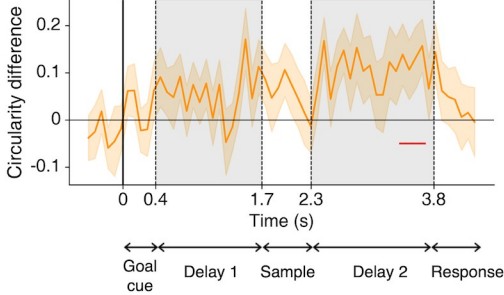

**D** Difference in posterior 2-D stimulus geometry between upper and lower half trials

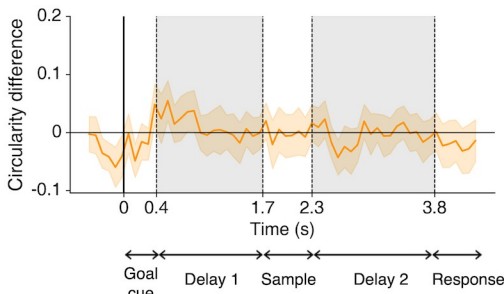

**Fig 3. Behavioral relevance of goal and stimulus geometry. (A)** Significant correlation (Spearman's *r*) between frontal 2D goal geometry in Delay 1 and the averaged response error. **(B)** Time-resolved difference in frontal 2D goal geometry between trials in the upper and lower quartiles based on response errors. Red horizontal lines denote significance time points at α = 0.05. **(C)** Same as **(B)** but for posterior goal geometry. **(D)** Difference in posterior 2D stimulus geometry using a median-split. Data and code that support these findings are available at: https://doi.org/10.57760/sciencedb.16868. 2D, two-dimensional.

bottom and top quartiles of each participant's trial-wise data based on the magnitude of combined response error to compute difference in individual circularity indices corresponding to the good and bad trials. Statistical significance was assessed by comparing the true values to a null distribution of circularity index differences generated via permutation (see Methods). In line with the correlation result, trials with smaller errors were associated with stronger 2D goal geometry than trials with larger errors. In frontal channels, this was true for Goal cue and Delay 1 (Fig 3B). For posterior activities, although we did not find a correlation with behavior, there was a significant difference in 2D goal geometry for part of Delay 2, temporally overlapping with the period of significant posterior 2D geometry (Fig 3C). We further demonstrated that worse circularity results for incorrect trials cannot be attributed to participants remembering the wrong goals (S3B Fig). Overall, these behavioral analyses support the functional relevance of the 2D goal geometry in working memory, suggesting that the degree to which individuals formed frontal task-congruent goal representations related to task performance. Interestingly, the timings during which the geometries manifested behavioral correlation diverged, coinciding with when frontal and posterior channels exhibited such representational structure, respectively.

## Posterior activities also exhibit 2D stimulus-specific geometry

In addition to the representations for task goals, what is the geometric structure for stimuli which themselves were drawn from a feature space consisting of 2 independent dimensions? Based on previous work using simple visual features that demonstrated low dimensionalities [34–37], it is expected that similar to goal representations, a task-congruent 2D subspace can well capture the stimulus-dependent variances in neural activity. When applying PCA to identify such subspaces specific to stimulus color and size, we observed the 2D stimulus geometry in posterior channels. The structure followed theoretical geometry (borders connecting neighboring coordinates did not cross each other) during sample presentation (Fig 4A and 4B), but not at other periods. In contrast, in frontal channels 2D stimulus structure was not observed throughout the trial, suggesting that stimulus-specific processing and maintenance were executed mainly by regions monitored by posterior channels. Of note, stimulus size and color were grouped into 3 bins each, resulting in 9 unique stimulus-specific conditions. In the current result, we used 6 of them in order to construct a close-formed geometric shape to calculate the circularity index (i.e., the middle row in Fig 1B, right was discarded).

Given that a 2D stimulus geometry was identified in posterior activities, it stands to examine whether such representations were related to behavior. No behavioral relevance was identified related to the strength of 2D stimulus representation during significant 2D representation (Sample) when we repeated the correlation ($r = 0.23$, $p = 0.29$) or split-trial analyses (Fig 3D).

Furthermore, to successfully adjust the sample features to the correct degree, participants might have preemptively manipulated the remembered stimulus in mind in preparation for the responses from Delay 2 onwards. To test this hypothesis, we performed the same PCA procedure but binned the trials based on the responded feature values and found that 2D response geometry was formed in posterior channels shortly after the onset of Delay 2, and similarly in frontal channels slightly later (Fig 4C and 4D). This possibly suggested that integration of goal and stimulus information indeed took place during goal implementation, leading to the formation of transformed stimulus (i.e., response) representation. It is notable that response representation exhibited poorer dissociation of size and color axes in group-level visualization (e.g., no linear planes that separate various sizes or colors can be found). One possible explanation is that response geometry is qualitatively worse as participants mentally generated the representation instead of viewing it physically [39]. Alternatively, this may suggest that the

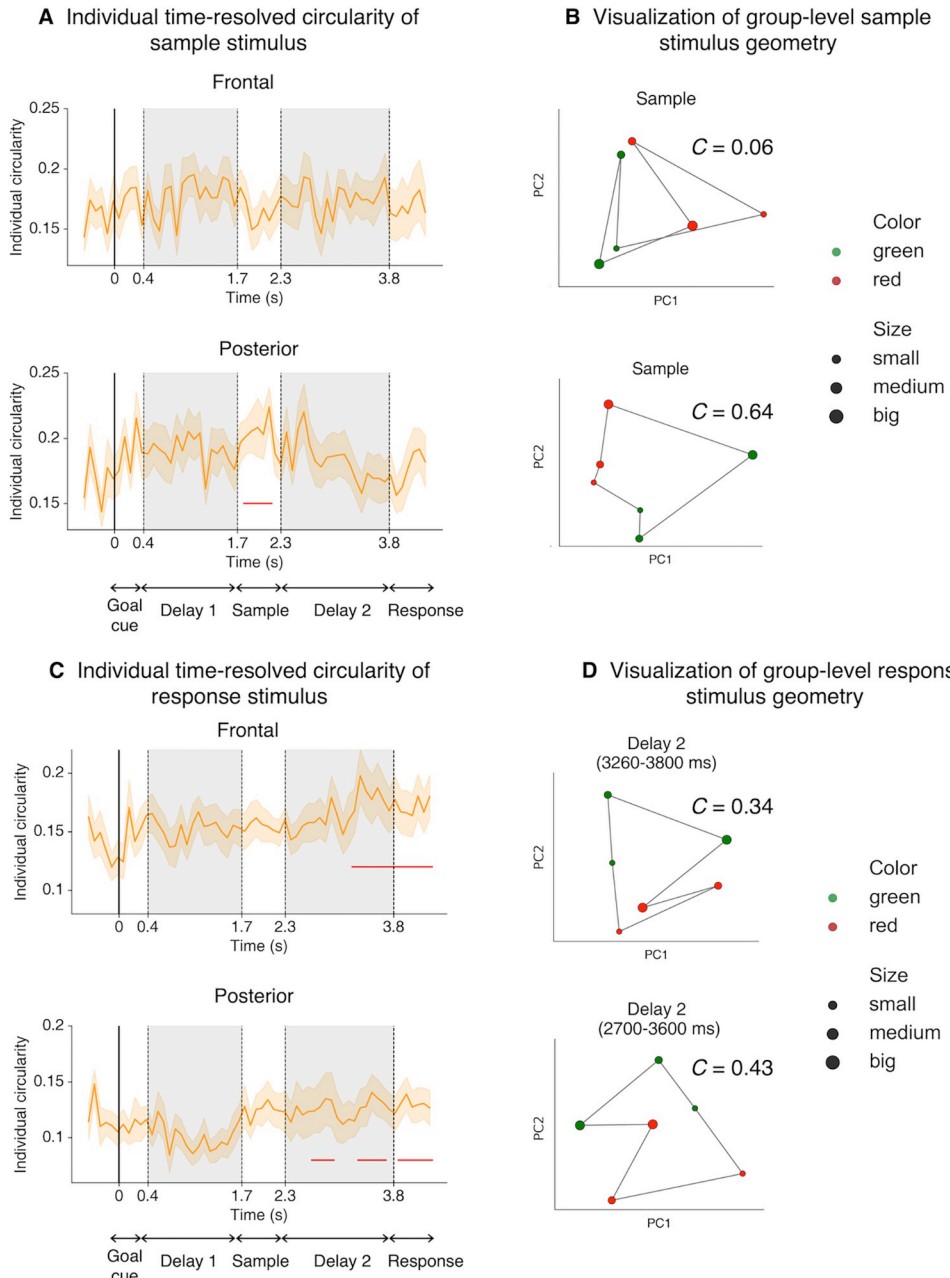

**Fig 4. 2D representational geometry of stimulus and response. (A)** Time courses of frontal (upper) and posterior (lower) individual circularity index using data averaged over a non-overlapping 80-ms sliding window. Error bars denote SEM. Red horizontal lines denote time points when 2D geometry was significant using a cluster-based permutation test ($\alpha$ = 0.05). **(B)** Visualization of group-level stimulus-specific 2D geometries for the Sample epoch. Stimulus feature values were aggregated into 3 bins and the middle color bin were discarded in order to calculate circularity index, resulting in 6 unique stimulus conditions. *C* denotes the circularity index. **(C)** Similar to **(A)** but for responded stimuli, which was calculated using the feature values after participant have finished adjusting (i.e., their answers). **(D)** Same as **(B)** but for response geometry. Data and code that support these findings are available at: https://doi.org/10.57760/sciencedb.16868. 2D, two-dimensional; SEM, standard error of the mean.

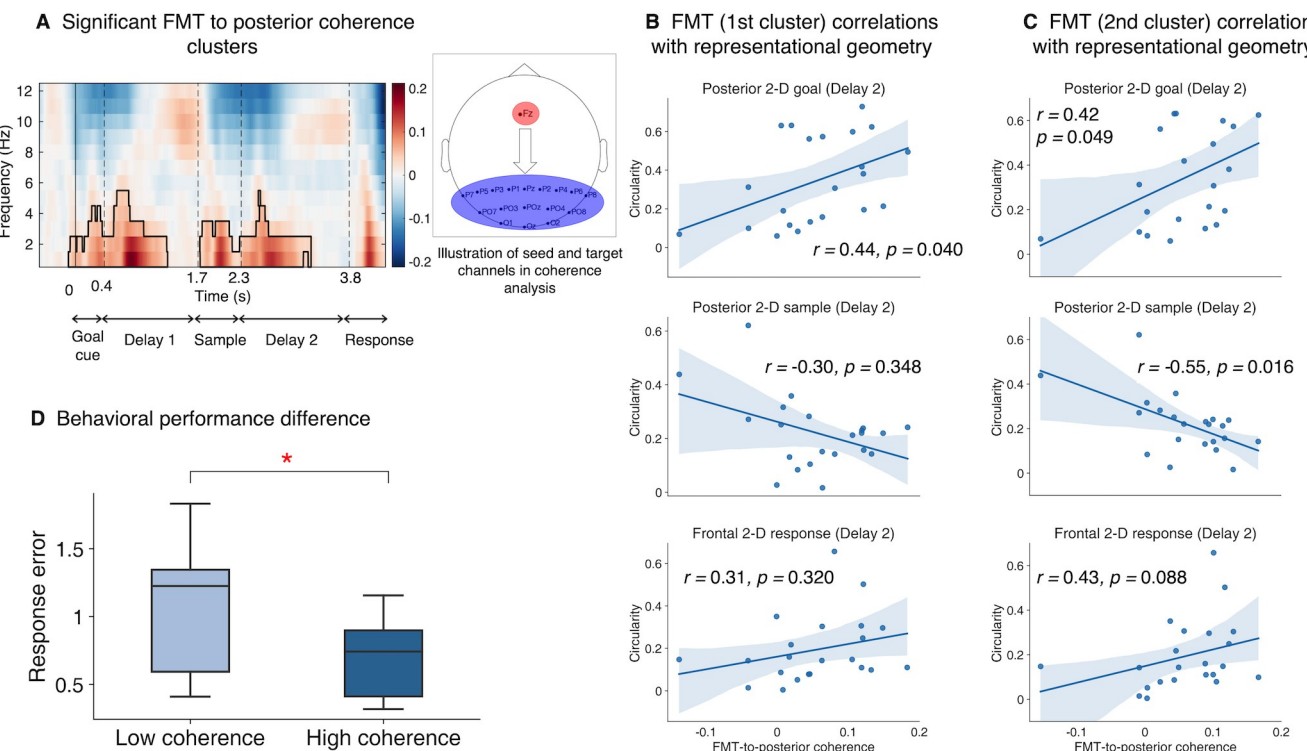

**Fig 5. FMT to posterior coherence and correlations with representational geometries. (A)** Time-resolved frontomedial (channel Fz)-to-posterior coherence between 1 and 12 Hz. Solid black line encircles the significant time-frequency clusters. For behavioral correlations, only subclusters falling within the 4 to 7 Hz (theta band) were used as masks to extract individual coherence strength. **(B)** Correlation results (Spearman's *r*) between the first FMT coherence cluster and representational geometries (i.e., circularity index) of task goals (upper), sample stimulus (middle), and response (lower) at Delay 2. Each dot represents an individual. Note that the spatial locations of significant correlations varied across the types of representation. **(C)** Same as **(B)** but using the second FMT coherence cluster. Correlation patterns were consistent between the 2 clusters, suggesting similar function of the FMT coherence at both times. **(D)** Difference in behavioral performance between individuals with low and high FMT coherence strengths in the second cluster. Data and code that support these findings are available at: https://doi.org/10.57760/sciencedb.16868. FMT, frontomedial theta.

response code across individuals might be variable so that no consistent axes could be detected by group-level PCA.

## Long-range theta connectivity mediates the backward transmission of 2D goal geometry and integration with stimulus information

So far, we have observed a distributed network for goal representations in which goal geometry transferred front to back in a task-congruent 2D format, and on top of that, task-congruent stimulus representations in posterior channels. However, what mechanisms mediate the backward transfer of 2D goal geometry and does this process relate to the integration of goal and stimulus information in working memory? Inspired by previous work demonstrating a prominent role of oscillatory coherence in long-range communication between brain regions [21,23–25], we targeted frontomedial theta (FMT) to posterior coherence as a measure of time-resolved long-range connectivity [40] and examined whether FMT was associated with the strength of neural geometries for task goals, stimulus, and responses. Firstly, 2 significant frontomedial-to-posterior coherence clusters (see Methods) were identified that covered the delta and theta frequencies (Fig 5A), one spanning from 0–1,276 ms (Goal Cue and Delay 1), the other from 1,728–3,268 ms (Sample and Delay 2). As the next step, we used the theta range (4 to 7 Hz) of these 2 group-level clusters as masks to extract individual FMT strength and examined its relationship with task representations and behaviors.

**Table 1. Correlations of theta coherence with goal- and stimulus-specific 2D geometries.**

| 2D geometry | Task epoch | Delay 1 theta coherence cluster | | Delay 2 theta coherence cluster | |
|---|---|---|---|---|---|
| | | r | p | r | p |
| **Posterior goal** | Goal cue | −0.094 | 0.661 | −0.321 | 0.925 |
| | Delay 1 | −0.008 | 0.661 | −0.118 | 0.925 |
| | Sample | 0.514 | 0.028* | 0.449 | 0.049* |
| | Delay 2 | 0.439 | 0.040* | 0.418 | 0.049* |
| **Posterior stimulus** | Goal cue | 0.075 | 0.665 | −0.054 | 0.538 |
| | Delay 1 | 0.097 | 0.665 | 0.045 | 0.580 |
| | Sample | −0.008 | 0.665 | −0.139 | 0.538 |
| | Delay 2 | −0.303 | 0.348 | −0.545 | 0.016* |
| **Frontal response** | Goal cue | −0.222 | 0.840 | −0.151 | 0.745 |
| | Delay 1 | −0.098 | 0.664 | 0.062 | 0.745 |
| | Sample | −0.151 | 0.840 | −0.132 | 0.745 |
| | Delay 2 | 0.311 | 0.320 | 0.430 | 0.088 |
| **Posterior response** | Goal cue | 0.377 | 0.114 | 0.146 | 0.455 |
| | Delay 1 | 0.226 | 0.206 | 0.073 | 0.455 |
| | Sample | 0.345 | 0.114 | 0.300 | 0.348 |
| | Delay 2 | 0.179 | 0.212 | 0.025 | 0.455 |

* Denotes $p < 0.05$, FDR-corrected.

2D, two-dimensional.

The first FMT cluster (Fig 5B, first row) was correlated with posterior 2D goal geometry at Sample and Delay 2; there was no significant result for such a relationship during Goal cue or Delay 1 (Spearman's rank correlation, one-tailed; Table 1). This suggested FMT to posterior connectivity could underlie the transfer of 2D goal representations to posterior sites which initially emerged frontally and provided a tentative explanation for the temporal cascading relationship observed earlier across frontal, central, and posterior channels. Moreover, the same effect was also found for the second FMT cluster (Fig 5C, first row), which could indicate both FMT clusters share the function of relaying the 2D goal information backwards.

We next wondered whether FMT was also related to the neural geometries of sample and response stimulus representations, as one potential function of long-range fronto-to-posterior connectivity is the transfer of task-related control signal to modulate stimulus processing. Based on this and our finding that FMT is involved in goal information relay, we hypothesized that FMT was related to the integration of goal and sample information, and therefore, the formation of response representations as the product. Specifically, FMT should be negatively correlated with sample, and positively correlated with response geometries, given that stronger goal representations should be associated with the rise of response representation and the decay of sample representations (due to transformation of original stimulus information). Aligned with the predictions, the second FMT cluster was negatively correlated with the 2D sample structure posteriorly at Delay 2 (Fig 5C, second row), although its correlation with posterior response structure was not significant. Nevertheless, recall that we observed evidence for response representations in both frontal and posterior channels, we thus examined the relationship between the second FMT cluster and response structure in frontal channels, and found the 2 were marginally positively correlated (Fig 5C, third row) at Delay 2. As a separate note, although significant delta coherence was also observed, it did not correlate with either neural or behavioral results, in contrast to theta-band activity (S1 Table).

Given the importance of transferring goal representations to influence stimulus processing in the present task, we investigated the relation between FMT connectivity and behavioral performance. Participants were median-split based on the FMT coherence strength and between-group difference in response error was tested. Those with stronger connectivity during Sample and Delay 2 (i.e., the second cluster) performed better overall, $t(20) = 2.17$, $p = 0.02$ (Fig 5D). On the contrary, there was no difference if participants were divided based on the first FMT cluster, $t(20) = 0.79$, $p = 0.22$. This distinction could be a result of the timing of the second FMT cluster being closer to the arrival of stimulus information and the implementation of task goals, hence, more linked with the underlying computations than the first one. Collectively, these findings illustrate that long-range connectivity is involved in the relay of goal information in a task-congruent format, which is in turn critical to the integration with the veridical stimulus to produce final responses.

## FMRI behavioral results

To further localize brain regions with the designed goal and stimulus geometries, we conducted an fMRI experiment using the same task ($n = 21$), with event timing adjusted to compensate for the sluggishness of BOLD signals. We first assessed participants' behavioral performance in the fMRI experiment following the behavioral measures in the EEG experiment. Overall, results from the fMRI experiment were comparable to those from the EEG experiment: mean size error was 4% of starting sample size (SD = 5%) and significantly different from 0 ($t(20) = 3.20$, $p = 0.004$). Mean color error was 1.76 steps (SD = 1.45) and larger than 0 ($t(20) = 5.59$, $p < 0.001$). Mean absolute size error was 14.9% of starting size (SD = 3.3%), while mean absolute color error was 22.7 steps (SD = 2.03).

## Whole-brain searchlight for task-congruent goal and stimulus geometries

In the EEG study, frontal and posterior channels displayed differential sensitivity to goal and stimulus representations. These signals likely arose from more anterior and posterior cortices, respectively. To have a finer understanding of the spatial localization of the 2D representational geometry associated with goal and stimulus space, we conducted individual searchlight analysis combined with circularity index to examine the 2D representational structures of goals and stimuli across the whole brain, using trial-wise beta coefficients estimated for each delay period from general linear models (GLMs).

We found comparable evidence for the 2D goal structure in frontal regions during Delay 1 and in posterior regions during Delay 2, confirming the results from our EEG experiment. Searchlight identified regions of interest (ROIs) mainly in the frontal cortex during Delay 1, including bilateral orbitofrontal cortex (OFC), inferior precentral sulcus (iPCS), right inferior frontal sulcus (IFS) (overlapping with traditional dorsolateral PFC) and gyrus (IFG), and left medial PFC (mPFC), in which neural activities showed the 2D structure predicted by the goal space (Fig 6A, left and Table 2). During Delay 2, while some frontal clusters remained, more were found to exhibit such goal-related geometry in posterior visual-related areas, including left middle temporal gyrus (MTG), early visual cortex (EVC), and lateral occipital area (LO), among other surrounding cortical regions in temporal and parietal lobes (Fig 6A, right and Table 2). These patterns were replicated using an alternative GLM method that can theoretically better separate delay activity from stimulus- and response-related activity (S4 Fig). Overall, during Delay 1, 2D goal geometry was mainly constrained to frontal regions, in line with our EEG finding; in contrast, there was a more distributed network during goal implementation where task goals were represented in a 2D format. Nevertheless, we demonstrated that more brain regions appeared in the posterior cortex by using the center MNI y coordinates

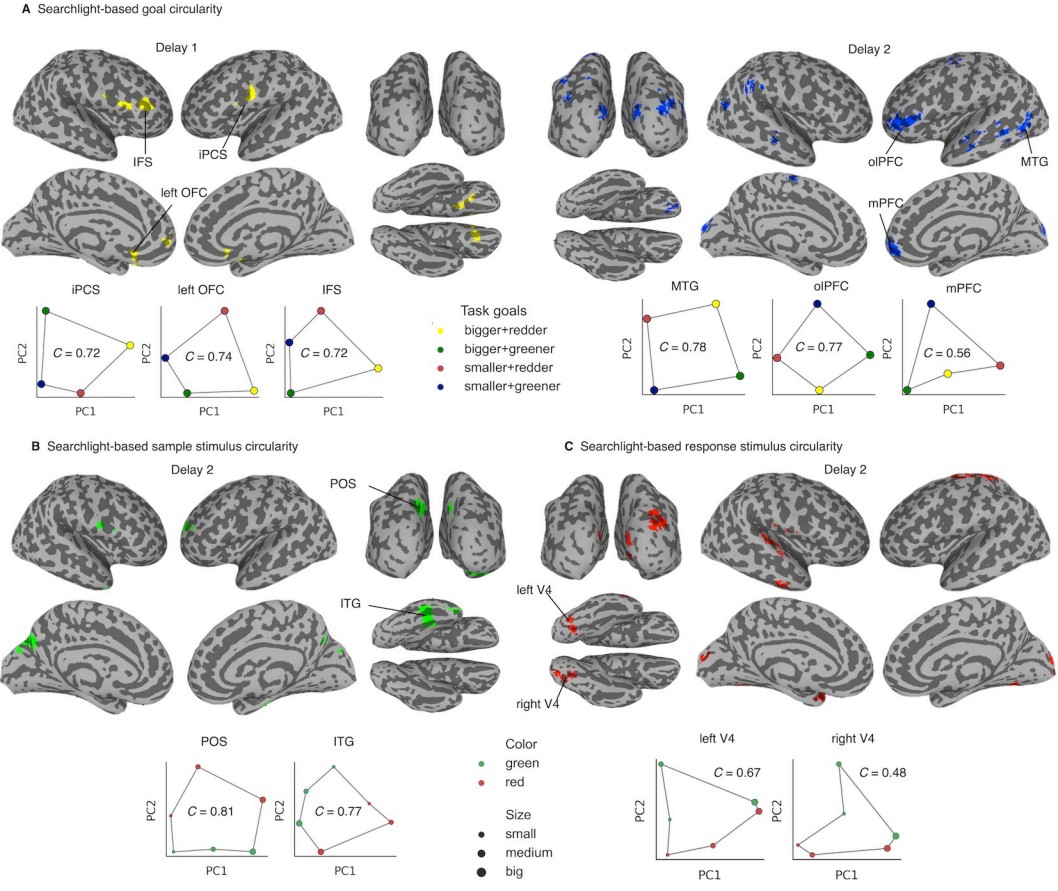

**Fig 6. FMRI whole-brain searchlight for goal, stimulus, and response geometries. (A)** Upper panel: Significant ROIs showing 2D goal geometry in Delay 1 (yellow) and 2 (blue). Lower panel: group-level visualization of the 3 largest clusters. **(B)** Significant ROIs showing 2D sample stimulus geometry in Delay 2 and group-level visualization of 2 example visual ROIs. **(C)** Same as **(B)** but for response stimulus geometry and group-level visualizations in Delay 2. An additional threshold of 50 voxels was applied to all statistical maps for display purposes. iPCS = inferior precentral sulcus; left OFC = left orbitofrontal cortex; IFS = inferior frontal sulcus; MTG = middle temporal gyrus; olPFC = orbital lateral prefrontal cortex; mPFC = medial prefrontal cortex; POS = parietal-occipital sulcus; ITG = inferior temporal gyrus; V4 = visual area 4. Data and code that support these findings are available at: https://doi.org/10.57760/sciencedb.16868. 2D, two-dimensional; fMRI, functional magnetic resonance imaging; ROI, region of interest.

(posterior to anterior axis) of all goal-specific clusters to conduct a two-sample rank-sum test for difference between the 2 delay periods. In line with the EEG experiment, the Delay 2 clusters lay more posterior to the Delay 1 clusters ($M_{Delay1} = 26.7$, $M_{Delay2} = -30.0$, $U = 86.0$, $p = 0.016$; S5 Fig). Furthermore, similar to the EEG study, we conducted RSA analyses using a whole-brain searchlight procedure and confirmed that the clusters with significant 2D geometry were largely replicated by the corresponding RSA model (S6C Fig).

On the other hand, 2D stimulus-specific geometry, as predicted by sample color and size, was identified most prominently in a visual cluster spanning left parietal-occipital sulcus (POS) and in higher-level visual processing area in inferior temporal gyrus (ITG; Fig 6B and S2 Table). This is comparable with evidence from posterior channels in EEG and in general consensus with previous findings that stimulus-related information was stored in corresponding sensory areas. For the frontal clusters in left middle frontal gyrus (MFG) and right precentral gyrus (PCG), they both exhibited poorer group-level results (Figs 6B and S7 for all ROIs).

**Table 2. Result summary of whole-brain searchlight for goal 2D geometry in Delay 1 and Delay 2.**

| Delay | Anatomical locations | Abbreviation | Cluster center MNI coordinates | | |
|---|---|---|---|---|---|
| | | | x | y | z |
| Delay 1 | Inferior precentral sulcus (left) | iPCS | −65 | 7 | 15 |
| | Orbitofrontal cortex (left) | left OFC | −15 | 19 | −27 |
| | Inferior frontal sulcus (right) | IFS | 36 | 40 | 3 |
| | Inferior frontal gyrus (right) | IFG | 60 | 17 | 9 |
| | Orbitofrontal cortex (right) | right OFC | 11 | 12 | −21 |
| | Medial prefrontal cortex (left) | mPFC | −12 | 55 | −1 |
| Delay 2 | Orbital lateral frontal cortex (left) | olPFC | −40 | 48 | 2 |
| | Medial prefrontal cortex (right) | mPFC | 6 | 61 | −4 |
| | Middle temporal gyrus (left) | MTG | −62 | −56 | 3 |
| | Early visual cortex (bilateral) | EVC | −1 | −92 | 17 |
| | SupraMarginal gyrus (right) | SMG | 65 | −33 | 34 |
| | Lateral occipital (right) | LO | 30 | −88 | 20 |
| | Superior parietal lobule (right) | SPL | 17 | −50 | 72 |
| | Inferior parietal lobule (left) | IPL | −23 | −55 | 65 |
| | Angular gyrus (left) | AG | −36 | −68 | 40 |
| | Superior precentral sulcus (left) | sPCS | −32 | −7 | 55 |
| | Ventral superior temporal sulcus (left) | vSTS | −60 | −32 | −5 |
| | Orbitofrontal cortex (right) | right OFC | 15 | 48 | −22 |
| | Posterior superior temporal sulcus (right) | pSTS | 44 | −58 | 15 |
| | Anterior superior temporal sulcus (left) | aSTS | −56 | −7 | −12 |

This observation was accompanied by small circularity values, in sharp contrast to the other visual-related regions with robust 2D geometry. One possibility is that group-level PCA required a consistent neural code across individuals in order to establish 2 principal axes that corresponded to the conceptual dimensions. In theory, this is more achievable in dedicated sensory areas where neurons are organized according to visual features, whereas in domain-general regions, the neural codes are likely to be non-sensory in nature (i.e., abstract code) and more variable across participants.

Lastly, given that the EEG experiment found 2D response representations in both frontal and posterior channels, which was modulated by FMT coherence marginally, we also sought to clarify this with a dedicated searchlight analysis. 2D response-specific geometry was found in visual areas including bilateral EVC, V4 and right visual area 3AB (V3AB), right superior temporal gyrus (STG) and right inferior temporal sulcus (ITS) as well as around central sulcus (CS; Fig 6C). Again, we note the same difference in group-level patterns between domain-specific versus domain-general regions (S7D Fig), in line with the proposition of the sensory and abstract coding divergence.

Taken together, data from the 2 experiments demonstrate the spatial and temporal characteristics of the 2D geometries are in fact robust and can be uncovered by different recording modalities.

### Functional connectivity between LPFC and posterior regions mediates 2D goal geometry and relates to task performance

Having identified regions exhibiting the 2D geometry related to task goals and sample stimuli, we evaluated whether long-range connectivity between these particular regions mediated the transfer of goal representations from Delay 1 to Delay 2, similar to what was observed in the

EEG experiment. We used trial-wise beta estimates of delay-period activities to compute functional connectivity among pairs of regions derived from the previous searchlight analyses. Given the large number of clusters, we opted for a data-driven approach of choosing the 3 largest clusters that showed significant 2D goal representation during each delay period. For stimulus- and response-specific regions, considering the potential distinction in coding schemes between the anterior domain-general (e.g., MFG and CS in S7 Fig) and posterior visual ROIs, only the largest clusters in visual-related regions were included. This additionally led to 2 posterior ROIs per searchlight being added into the functional connectivity analyses, bringing the total to 10 seed regions.

In light of the EEG connectivity result that long-range connectivity mediated the relay of goal information, a similar approach in the fMRI domain was employed. The advantage of fMRI allowed us to further limit this test to ROIs specifically holding the 2D goal or stimulus/ response geometry: for Delay 1, 3 clusters iPCS, left OFC, and IFS were chosen (yellow nodes in Fig 7A); for Delay 2, the top 3 clusters in size were MTG, olPFC, and mPFC (blue nodes in Fig 7A). Lastly, 2 posterior seed regions (green nodes in Fig 7A) with 2D stimulus-specific geometry in POS and ITG, and 2 seeds (red nodes) with 2D response-specific geometry in left and right V4 were included.

Firstly, we assessed whether functional connectivity between Delay 1 and Delay 2 goal-related seed regions was associated with the strength of 2D goal geometry in Delay 2 clusters. The connection between iPCS and MTG at Delay 2 was positively correlated with 2D goal-geometry in the latter (Fig 7C; Spearman correlation; $r = 0.53$, $p = 0.041$, BF10 = 11.48). Additionally, another Delay 1 region IFS also showed a similar relation with the same Delay 2 seed

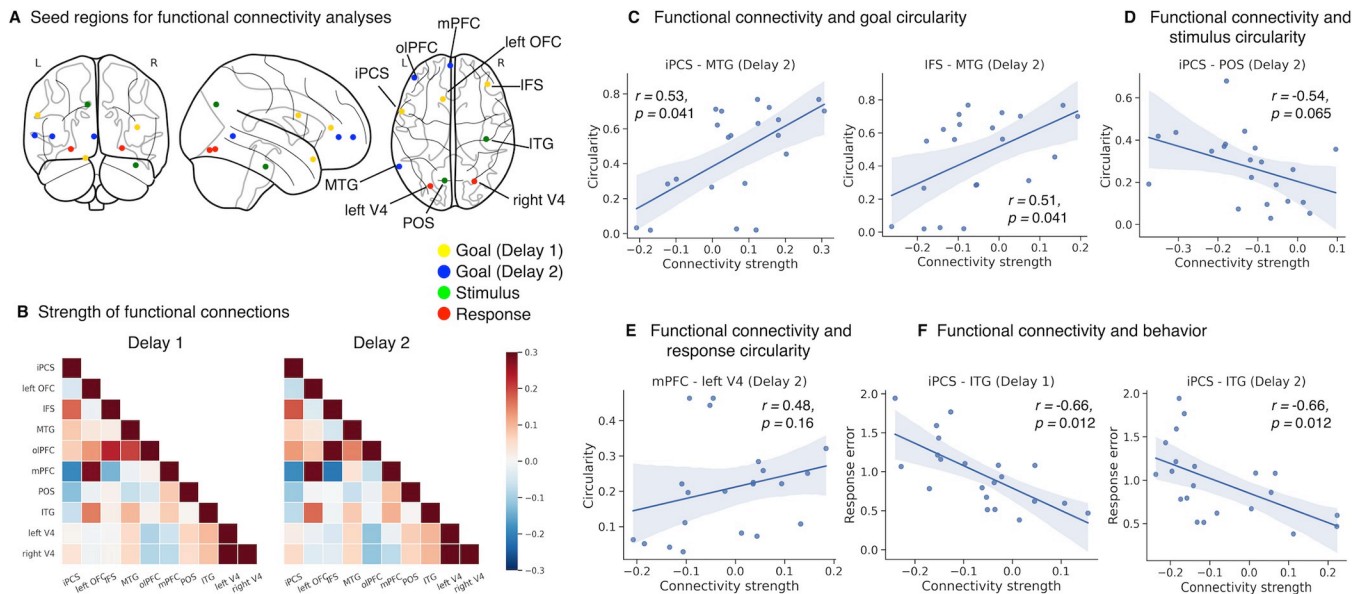

**Fig 7. Functional connectivity among selected ROIs and correlations with neural and behavioral measures. (A)** Illustration of seed region locations. Yellow and blue nodes represent ROIs showing 2D goal geometry in Delay 1 and 2, respectively. Green and red nodes represent ROIs showing 2D stimulus and response geometry, respectively. **(B)** Matrices showing strength of pairwise connections among seed regions. **(C)** The connection strength between iPCS-MTG and IFS-MTG in Delay 2 was associated with goal circularities within MTG. **(D)** Higher connection strength between iPCS (Delay 1 goal) and POS (stimulus) in Delay 1 was associated with lower 2D stimulus geometry in Delay 2. **(E)** Higher connection strength between mPFC (Delay 2 goal) and left V4 (response) in Delay 2 marginally correlated with higher 2D response geometry in Delay 2. **(F)** Stronger functional connectivity between iPCS (Delay 1 goal) and ITG (stimulus) in both delays was correlated with higher performance. iPCS = inferior precentral sulcus; left OFC = left orbitofrontal cortex; IFS = inferior frontal sulcus; MTG = middle temporal gyrus; olPFC = orbital lateral prefrontal cortex; mPFC = medial prefrontal cortex; POS = parietal-occipital sulcus; ITG = inferior temporal gyrus; V4 = visual area 4. Data and code that support these findings are available at: https://doi.org/10.57760/sciencedb.16868. 2D, two-dimensional; ROI, region of interest.

($r = 0.51$, $p = 0.041$, BF10 = 7.21). These findings pointed to the MTG as a key target region for potentially receiving 2D goal information from lateral frontal cortex. It is also noteworthy that although the searchlight identified frontal ROIs in Delay 2, seemingly contradictory to the EEG result, goal geometry within these clusters (i.e., olPFC and mPFC) was not mediated by connections with Delay 1 regions (S3 Table), suggesting they may be separate from the process of relaying goal information.

Moreover, we also tested the relations between functional connection and the original and response stimulus circularity. Specifically, all pairs between goal-specific and stimulus- or response-specific regions were included in these analyses. We found a trend suggesting similar patterns with EEG: the Delay 1 iPCS seed marginally modulated stimulus geometry in POS, with stronger connections accompanied by weaker 2D stimulus representation ($r = -0.54$, $p = 0.005$, uncorrected; or $p = 0.065$, FDR-corrected; BF10 = 11.28; Fig 7D). The opposite trend was identified for a Delay 2 goal-related seed (mPFC) and a response-related seed (left V4), indicating stronger 2D response representation in V4 with stronger connections between the 2 ($r = 0.48$, $p = 0.013$, uncorrected or $p = 0.160$, FDR-corrected; BF10 = 5.33; Fig 7E).

Finally, we sought to examine whether any functional connectivity could relate to behavioral performance. Interestingly, all pairwise connections that showed a significant relationship with response error before multiple comparison correction were between Delay 1 goal-specific and stimulus-specific regions (S4 Table), demonstrating an extremely consistent pattern. Among them, iPCS (Delay 1 goal-specific seed) and ITG (stimulus-specific seed) in both delays remained significant after FDR correction ($r = -0.66$, $p = 0.012$, BF10 = 74.14; $r = -0.66$, $p = 0.012$, BF10 = 76.11; Fig 7F).

In short, the fMRI connectivity analyses replicated the findings of the EEG experiment: long-range connectivity between frontal and posterior cortices was associated with the backward transmission of 2D goal geometry and the transformation of stimulus information, which subsequently influenced behavior.

## Discussion

In the present study, we investigated the distributed goal representations in working memory, using a newly developed behavioral paradigm that required retention of both task goals and stimulus contents in designed 2D goal and stimulus spaces. Leveraging recently advanced state space analyses, our first EEG experiment revealed that the representational geometry of goals followed the theoretical 2D structure. This 2D goal geometry first emerged in frontal channels during goal maintenance and then transferred to posterior channels for goal implementation. Notably, the fidelity of the goal geometry was associated with individual memory performance. Meanwhile, a 2D stimulus geometry was observed in posterior channels in accordance with the theoretical stimulus structure. The frontal goal geometry transferred to posterior sites and interacted with the posterior stimulus geometries through FMT coherence. A second fMRI experiment further indicated that both LPFC and OMPFC exhibited the desired 2D goal geometry. However, only LPFC demonstrated significant functional coupling with posterior visual-related regions that related to the transfer of goal structure and behavior. Collectively, our findings suggest a potential neural mechanism for how frontal and posterior cortices are orchestrated through communications of task-congruent geometry to implement computations necessary for goal-directed behavior.

### Task-congruent representations for transferring task information

Frontal cortex has long been considered as a core region for processing abstract task information, with aggregated BOLD activity that tracks levels of abstraction during cognitive control

[11,12], and successful decoding of a variety of abstract task information [16,17,41,42]. Beyond the focus of frontal areas, many studies have also reported successful decoding of task sets in targeted sensory cortex [16,19]. We directly addressed the roles of the distributed representations of abstract task information, and how they support goal-directed behavior in working memory by tracking the representational geometries of structured task information in the brain. In particular, we designed a structured 2D task goal space, incorporated with a working memory manipulation task in a 2D stimulus space, to systematically investigate the corresponding task representations. Compared with previous work, this design enabled the characterization of the representational geometry of multiple task variables (goal, stimulus, and response), their coding dynamics, and their interactions within a single paradigm. During the first delay when goals were encoded but not yet implemented, task goals awaiting to be integrated with specific contents in working memory were indeed maintained in a compressed format of representations that reflected the designed dimensions. Importantly, this task-congruent goal geometry was associated with subsequent memory performance and cannot be accounted for by preparative motor planning signals. These together suggest that at the population level, frontal cortex represents abstract goal information in a manner similar to how sensory cortex encodes feature information [33,34,37], despite neurons in the 2 cortices possess distinct tuning properties (i.e., mixed versus simple selectivity). In other words, frontal cortex organizes goals into an abstract relational space based on their underlying structure, which can be maintained in working memory for later control of stimulus adjustments [43]. This is reminiscent of studies on PFC demonstrating relational structures of task knowledge, such as inferred latent state [44,45], schema [46] and indirectly observed associations between stimuli [47,48].

When moving onto the second delay period during which participants needed to implement the maintained goal on specific memory contents, frontal cortex no longer retained the task-congruent goal geometry, at least not in an active format, perhaps due to that frontal 2D goal geometry was primarily involved in goal maintenance rather than goal implementation. Conversely, the 2D goal geometry gradually developed in central cortex and later in posterior cortex, indicating that the task-congruent geometric information was communicated to sensory cortex for goal implementation. Because population-level neural geometry can remain stable when underlying single-neuron activities change [38], one benefit of forming task-congruent, low-dimensional geometry of task information could be that the stable geometry facilitates information relay between cortical regions through communication subspaces [28,29]. This shift in locations was accompanied by a correlation of behavioral performance with posterior goal representations, reminiscent of previous findings showing a relationship between decoding accuracies in visual areas and reaction times using a different task design [16]. Overall, our results are comparable with a growing body of evidence that supports a distributed network for abstract task information during active implementation [19,20], and further extend the idea by proposing that the task information originates from communications between frontal and posterior regions in a task-congruent format.

In parallel, a similar 2D stimulus geometry in which color (red-greenness) and size were represented as dissociable dimensions was only found in posterior activities during sample presentation and the ensuing delay. This result lends more credence to our PCA-based approach and aligns with the sensory recruitment account of working memory, which proposes that the storage of sensory stimuli relies on the same visual regions that initially encode the information [2,3,10]. Besides stimulus-related processing, dedicated sensory regions is also likely to be the locus of goal and stimulus integration, as implementation of task goal and its functional relevance also primarily involved posterior activities. Notably, the fact that we observed task-congruent stimulus representation in posterior cortex does not necessarily

imply that representation in this region is always low-dimensional. In fact, the dimensionality of sensory representation could well depend on the nature of the stimuli, such as the number of relevant features. For example, when naturalistic objects or pictures were used, the neural responses in the visual cortex can be high-dimensional [49]. By contrast, when structured stimuli were used (such as orientations or colors spanning a circular space), the representational geometry in the visual cortex can exhibit circular patterns and hence low-dimensional, meaning that only 2 primary dimensions are needed to explain sufficient variance in the neural data [33,37]. Given that our study artificially defined 2 stimulus axes and participants repeatedly learned about this space, it is possible that the stimulus representations reflected the color and size dimensions, with all the other irrelevant dimensions (e.g., stimulus category, texture) being compressed. We did not test the stimulus dimensionality in posterior cortex prior to task learning. Therefore, whether stimulus dimensionality in our study was reduced after learning remains to be tested in future research.

While the task-congruent, 2D goal representations were present and contributed to memory performance, we acknowledge that other formats can coexist. Whereas low-dimensional task representations reflecting constituent dimensions support stability in neuronal readouts, knowledge generalization and efficient learning [26,27], high-dimensional task representations endow high separability between representations and facilitate cognitive flexibility [50]. In fact, using a model that assumes all goals are equidistant without forming any structure, we indeed observed such a conjunctive representational format at different times and/or in different brain regions compared to the 2D representations. For example, significant conjunctive representation was predominantly observed in the posterior channels of EEG from the presentation of goal cue onward, suggesting that it likely reflected cue-driven, sensory signals. This pattern, however, was less evident in the fMRI study, possibly because the cue was presented in a textual format (S6D Fig). Moreover, we note that the 2D geometry is highly tied to the experimental design in the current study. Future studies that include a task with a goal structure of different dimensionality as well as investigate the functions of alternative coding formats will add to a more comprehensive view of the representational geometry of abstract task variables [27,50–52].

It is noteworthy that applying state-space methods to EEG/fMRI data has inherent limitations. The application is based on the assumption that, while each voxel/electrode represents aggregated activity from many neurons, the collective preference of these neurons may manifest at the single-voxel/electrode level, giving rise to significant geometry patterns at the population level. However, it is also plausible that brain regions without significant goal or stimulus geometries according to our analyses may in fact maintains such geometries, but at a resolution undetectable by EEG or fMRI. This remains an issue for most (if not all) EEG/fMRI studies, and future work with finer spatial resolution may provide deeper insights into this issue.

### Interaction between goal and stimulus representations in the form of task-congruent geometry

What mechanisms mediate the transfer of task goals and support the necessary integration of goal and stimulus information? We proposed FMT-to-posterior coherence as a candidate through which higher-order task information is transferred to the locus of specific content storage. More importantly, the transmission retains the geometrical property of the representations, manifested by the relationship between the strength of the connectivity and goal circularity. Although FMT has long been linked to top-down processes during working memory [40,53,54], its functional interpretations are not straightforward [55], ranging from coordinating reactivation of working memory items [24,40], gating of working memory encoding and

maintenance [56], to prioritizing internal representation [57]. The evidence provided here points to a more specific role in mediating the communication between frontal and posterior areas for goal representations and the subsequent integration with stimulus representations, while embedded within the general notion that frontal theta oscillation exerts cognitive control signals by synchronizing activity of task-relevant information [58,59]. Furthermore, functional coupling-dependent transmission of goal representations was corroborated by the fMRI connectivity results, involving regions specifically maintained task goals with the matching geometric format. While functional connectomes measured by EEG phase-based metrics and fMRI overlap only to a moderate degree (approximately 0.4) [60], caution is warranted when attributing the cross-modal results to the same underlying neural generator. We nonetheless provide converging support for a fronto-posterior connectivity related to the transmission of 2D goal representations. Notably, the relationship between FMT and goal geometry was only correlational in the current study and how this interaction takes place at the mechanistic level remains unclear. Recent theoretical work proposes a mechanism by which oscillatory signals can interact with neuronal spike timing within one brain region to encode stimulus-specific information in a phase-dependent manner [21]. Its potential relationship with the current result remains to be further tested.

The 2 experiments also uncovered an interesting dynamic between the representations of remembered and transformed stimuli. In the EEG study, transformed stimulus representations (i.e., participant's responses) were formed during Delay 2 and were accompanied by a decrease in the strength of original stimulus representations in the posterior channels. The simultaneous rise of response representation and fall of sample representation, both modulated by fronto-to-posterior coherence, allude to the possibility that stimulus transformation depends on the integration of task goals, and participants might have reduced or even discarded the original copy of remembered items to mitigate interference [39]. In line with this finding, in the fMRI study, representations of both original and transformed stimuli were observed in visual-related regions during Delay 2, albeit in different subregions. Moreover, 2D stimulus and response representations were also observed in more anterior cortex, but in a manner distinct from visual-related regions as demonstrated by the group-level visualization results. Specifically, we argue that the visual-related regions might recruit a more aligned neural code across participants for representing stimulus information, while the way in which brain regions beyond visual regions represent stimulus information can be more variable and sensory independent. This would be consistent with previous studies demonstrating an orthogonal common "template" subspace in PFC ready for response, in contrast to choice-invariant stimulus representations in visual cortex [35]. The 2D response representation in the anterior cortex might have reflected a similar neural subspace dedicated to guiding behavior. In addition, since our study specifically tested for the task-congruent format which was shown to be functionally relevant to behavior, it is entirely possible that original and transformed stimuli were encoded with lower precision or in a different format [61,62] within specific modalities or ROIs that were not detectable using current methods.

## Differential functions of LPFC and OMPFC

The limited spatial resolution of EEG has precluded precise localization of the 2D goal geometry in frontal cortex. With the follow-up fMRI experiment, we further localized the 2D goal geometry to subregions within LPFC and OMPFC. Specifically, 2 LPFC clusters, the IFS and iPCS, demonstrated significant 2D goal geometry, of which the functional connectivity with posterior regions related to the strength of goal geometry in posterior ROIs as well as memory performance. In contrast, the relationships were largely absent in MPFC and OFC clusters.

This functional distinction between LPFC and OMPFC aligns with recent theoretical work on the differential roles of task representations in LPFC and OMPFC, purporting that the LPFC uses task representations to build rules for action selection, whereas the OMPFC abstracted task knowledge in a relational map [43]. In line with this notion, only LPFC is actively involved in the coordination between frontal and posterior cortex which likely supports the transmission of goal geometry for implementation. Our newly designed paradigm offers a useful tool for investigating the functional distinction between the 2 networks. Future work with a more comprehensive examination on the neural geometries in subregions of the 2 networks is needed to further address this question.

## Conclusions

In summary, across 2 experiments, we provided converging evidence for how distributed task representations emerge and transfer in working memory to support goal-directed behaviors. In particular, frontal cortex maintains task-congruent neural geometries of goal representations in preparation for subsequent goal implementation in posterior visual-related cortex. Our findings highlight working memory as a multicomponent and collaborative cognitive system that relies on coordinated neural interactions across multiple brain regions.

## Materials and methods

### Participants

Twenty-three participants were recruited for the EEG experiment (mean age = 24.3 years; age range = 19 to 30 years; 14 females), one was excluded due to excessive noise (over 20% of total trials). Twenty-one MRI-eligible participants were recruited for the fMRI experiment (mean age = 24.2 years; age range = 21 to 30 years; 16 females). There is no overlap in participants between studies. All participants were recruited from the Shanghai Institutes for Biological Sciences community, reported neurologically and psychologically healthy, had normal or corrected-to-normal vision, provided written informed consent, and were monetarily compensated for their participation. The study was approved by the Ethics Committee of the Center for Excellence in Brain Science and Intelligence Technology, Chinese Academy of Sciences (CEBSIT-2020028) and conducted according to the principles expressed in the Declaration of Helsinki.

### Experimental design and procedure

**Overview.** We designed an experimental task that required the maintenance of both a goal and a specific stimulus in working memory and at later stage the manipulation of the stimulus features based on the goal. Specifically, each remembered stimulus varied along 2 orthogonal stimulus dimensions, size (small to big) and color (green to red). Correspondingly, there were totally 4 types of goals composited by 2 orthogonal dimensions of size and color manipulations: adjusting the remembered stimulus to be **(1)** bigger and redder; **(2)** bigger and greener; **(3)** smaller and redder; and **(4)** smaller and greener. The 2D stimulus space, as well as the extent of required adjustment in size and color were predefined and learned in a behavioral session preceding the main task.

### Definition and manipulation of stimulus features

Color (green-redness) were adjusted by first converting the images into grayscale to remove all original hues while preserving the opacities of pixels using the Image module from Python package PIL. Then, the resulting files were converted to RGBA mode again and the green-

redness values were manipulated by adding/subtracting the same value in the Red and Green channels, whereas the Blue channel was set to zero. This made so that the stimulus transitioned from extremely green to extremely red in a smooth fashion (for illustration, see Fig 1D). The acceptable RGB value range for PsychoPy was 0 to 1, each button press moved the R and G value of all image pixels to opposite directions by 0.01. The values in R and G channels were rescaled to 0.25 to 0.75 with a mean of 0.5. Therefore, for each stimulus there existed 150 variations of color-manipulated images (until all pixels had a value of 0 or 1). However, since all pixels appearing uniformly green or red would render the stimulus unidentifiable (because there is no contrast), the 30 images with most extreme R/G values were discarded, resulting in 120 steps for the range of color adjustment. Both starting size and colors were drawn from 3 predetermined bins, resulting in 9 unique conditions, which were used for subsequent analyses. For size the bins were 0.17 ± 0.01, 0.22 ± 0.01, or 0.27 ± 0.01 of screen height; for color they were 34 ± 2, 58 ± 2, and 82 ± 2, indexing from the 120 color steps. Stimulus size was directly controlled in PsychoPy program using the size attribute and the allowed range was set to 0.01 to 0.45 screen height. The distance between each particular sample stimulus and its target answer for size was ±24% of the original size (unit: screen height) and ±26 (unit: color steps) for color. Participants learned the required distance during a behavioral learning session.

## Behavioral learning

In the behavioral session, which was held 1 or 2 days before the main task session, participants learned the degree of required adjustment by first viewing all pairs of starting and target values in size and color shown side-by-side, respectively (162 trials per stimulus feature), before receiving a Two-Alternative-Forced-Choice (2-AFC) test whereby they were given a starting stimulus and asked to choose the correct target stimulus from 2 options (80 test trials per stimulus feature). Next, both features were combined together in the same stimulus in another round of 2-AFC test to familiarize them for simultaneous adjustment of both size and color based one of the 4 goals, in preparation for the main working memory task (100 trials). Finally, participants completed 180 trials for the main task in order to apply the learned degree of adjustment on stimulus features. In the EEG experiment, the 4 goals were associated respectively with different shape cues, which the participants also learned and practiced the association between shape cues (circle, square, triangle, and pentagon; Fig 1A) and goals in these 180 trials. In the behavioral session only, trial-wise feedback of response error was given so that participants could keep improving their performance.

## EEG main task

The main task consisted of 2 memory delays and a response period. The first delay required the maintenance of the goal only, and the second delay required both the maintenance of the goal and the stimulus. At the beginning of a trial, there was a 300-ms fixation period followed by one of the shape cues for 400 ms appearing centrally (Goal cue), signaling the direction of manipulation in the current trial. This was followed by a 1,300-ms delay (Delay 1), during which participants should keep maintaining the manipulation goal. Following the first delay, a stimulus was presented for 600 ms (Sample). The stimuli were randomly chosen from the exemplars belonging to three different conceptual categories with similar shapes: bowling pins, plants, and microphones, provided and detailed in [63]. Participants were instructed to maintain the sample's size and color during a second delay period for 1,500 ms (Delay 2). To prevent participants from using physical changes on the retinal image as memory aids, the stimulus was presented in a random location of an invisible circle around the center of the

screen with a radius of 0.08 screen height, making it a 2˚ visual angle difference from the center. During the response period (Response), the same stimulus reappeared with a randomly chosen but different size and color from the remembered values to prevent response preparation. Participants were given a maximum of 6,000 ms to adjust the features of stimulus on screen to the correct degree using button presses. The correct response depended on both the starting values and the cued goal, as learned in the behavioral session. Thereafter, a variable intertrial interval was incurred (800 to 1,200 ms, drawn from a uniformed distribution). Overall, participants completed 648 trials divided into 12 blocks lasting for approximately 2 h, with task variables of goal types and stimulus features counterbalanced, resulting in 18 trials in each of the possible combinations. Of note, during the EEG recording participants only received averaged feedback on their performance after a block, to avoid further improvement on the extent of change per se as a result of trial-wise feedback.

### fMRI main task

Behavioral paradigm and experimental procedure in the fMRI experiment generally remained consistent with the EEG study unless otherwise stated. In-scanner task shared the same components as in EEG but with adjusted length to accommodate the delay in hemodynamic response function (Fig 1E, bottom). The durations of epochs were as follows: Goal cue = 500 ms; Delay 1 = 5,500 ms; Sample = 600 ms; Delay 2 = 8,400 ms; Response = 5,000 ms. Intertrial intervals were randomly chosen from 4,000, 5,500, and 7,000 ms with equal likelihood. Of note, goal cue was presented as texts instead of associated shapes in the fMRI task; therefore, participants were no longer required to learn the pairings. In total participants completed 12 functional blocks, each containing 18 trials and lasting 466.5 s.

### EEG apparatus

Stimulus presentation was implemented using PsychoPy (version 2021.2.3) [64] on a 48 × 27 cm HIKVISION LCD screen with a 60 Hz refresh rate and a 1,920 × 1,080 resolution. Stimuli were shown in white font on a gray background (RGB = 128, 128, 128) at a distance of 62 cm. During the task, head position was stabilized by a chin rest. Responses were given with the right hand on the 4 arrow keys on a keyboard. EEG data were recorded using a Brain Products ActiCHamp recording system and BrainVision Recorder Software (Brain Products GmbH, Gilching, Germany). Scalp voltage was obtained from a broad set of 59 channels at 1,000 Hz according to the extended 10 to 20 positioning system (FCz as reference). Channel impedance was kept below 20 kΩ.

### EEG preprocessing

EEG data were preprocessed in MNE-Python [65], which firstly involved down-sampling to 250 Hz and bandpass filtering using a high-pass filter of 0.1 Hz and a low-pass filter of 40 Hz. The continuous raw data were then segmented into epochs, corresponding to 500 ms before the onset of the goal cue (200 ms before the fixation onset) until 600 ms after the onset of the response period. EEG channels with excessive noise were identified through visual inspection and replaced via interpolation using a weighted average of the surrounding channels. Each epoch was inspected visually for artifacts such as excessive muscle movements and amplifier saturation, and contaminated trials were discarded. Stereotyped artifacts such as ocular movements were subsequently removed from the data via independent component analysis. The data were baseline corrected for the subsequent analyses using signals from the time window of −200 to 0 ms before fixation onset. There were on average 621 trials per person remained after epoch rejection (SD = 26 trials).

## fMRI data acquisition

MRI scanning was performed at the Functional Brain Imaging Platform (FBIP), Center for Excellence in Brain Science and Intelligence Technology, Chinese Academy of Sciences (CEBSIT, CAS), on a Siemens 3T Tim Trio MRI scanner with a 32-channel head coil. High-resolution T1-weighted anatomical images were acquired using a magnetization-prepared rapid gradient-echo (MPRAGE) sequence (2,300 ms time of repetition (TR), 3 ms time of echo (TE), 9° flip angle (FA), 256 × 256 matrix, 192 sequential sagittal slices, 1 mm$^3$ isotropic voxel size). Whole-brain functional images were acquired using a multiband 2D gradient-echo echo-planar (MB2D GE-EPI) sequence with a multiband acceleration factor of 2, 1,500 ms TR, 30 ms TE, 60° FA within a 74 × 74 matrix (46 axial slices, 3 mm$^3$ isotropic voxel size). After 4 experimental runs we also acquired whole-brain Fieldmap phase and magnitude images for correction of EPI distortions. Stimuli were presented on a 1,280 × 1,024 resolution MRI-compatible screen at the back of the scanner, and participants viewed the screen through a mirror attached to the head coil with a viewing distance of 90.5 cm. They used 2 two-button response boxes, one in each hand to adjust the stimuli.

## fMRI preprocessing

Preprocessing of MRI data was performed using fMRIPrep 21.0.2 [66], which is based on Nipype 1.6.1 [67]. For each experimental run, the single-band reference data were taken as the reference volume. A B0 nonuniformity fieldmap was estimated and aligned to the reference volume. The reference volume was corrected for distortions using the fieldmap and was co-registered to the anatomical scan. Both the anatomical and functional scans were then normalized to the MNI152 template.

## Quantification and statistical analyses

**EEG: Neural geometries of task representations.** To uncover the geometry of neural representations associated with goals, sample stimuli, and responses, and to describe their similarities to goal or stimulus space, we conducted PCA to identify task-congruent subspaces in which the relevant task variables were encoded. Specifically, each participant's voltage data was collapsed over trials from the same conditions (goals, stimulus, or response bins), with each column standardized independently before applying PCA, resulting in a matrix of shape n_conditions × n_channels. Note that we used the same standard to divide sample stimuli and responses into bins, as the latter are essentially manipulated stimuli that share the same feature value ranges as the former. We repeated this procedure for different sets of channels (frontal, central, or posterior channels) separately. Frontal channels include: Fp1, Fz, F3, F7, F4, F8, Fp2, AF7, AF3, AFz, F1, F5, F6, AF8, AF4, F2; posterior channels include: Pz, P3, P7, O1, Oz, O2, P4, P8, P1, P5, PO7, PO3, POz, PO4, PO8, P6, P2; central channels include: FC5, FC1, C3, T7, CP5, CP1, CP6, CP2, Cz, C4, T8, FC6, FC2, FT7, FC3, C1, C5, TP7, CP3, CPz, CP4, TP8, C6, C2, FC4, FT8.

Unless stated otherwise, the PCA-related analyses and ensuing behavioral correlations were based on PCA performed on these individual data matrices. For the group-based visualization of the subspace, the individual averaged activity patterns were subsequently concatenated horizontally, resulting in a matrix of shape n_conditions × (n_channels × n_participants). All remaining steps followed the aforementioned procedure for individual data. Depending on the specific analysis, the procedure was either applied to individual time points or within each temporal epoch of a trial (i.e., Goal cue: 0 to 400 ms; Delay 1: 400 to 1,700 ms; Sample: 1,700 to 2,300 ms; Delay 2: 2,300 to 3,800 ms; and Response (3,800 to 4,300 ms) during which the data was averaged.

**EEG: Circularity index.**   To quantitively assess the structure of representations in the space reconstructed by PCA in spite of differences between task variables, individuals or principal planes, we used a simple metric, namely circularity index to capture the spatial properties of the neural geometry. Briefly put, circularity [68] of a shape was defined as the ratio between its area and the square of the total length of perimeter:

$$Circularity = 4\pi \times \frac{Area}{Perimeter^2}$$

A circle would always have a circularity of 1. If goal representations abided the relational structure formed in the hypothetical goal space consisting of 2 orthogonal dimensions (goal size and goal color) with equal distance between goals, the circularity would be that of a square, which is approximately 0.78. Since there were only 4 points to define the structure, this value was also the maximum as any quadrilateral would have a smaller circularity. However, in real data the magnitude of the circularity values could depend on several factors, including the number of conditions and number of participants being averaged (in the individual circularity analysis). To calculate circularity in the neural space, borders were drawn between the projected coordinates of each unique conditions, in the same order as in the hypothetical space (Fig 1B). For example, for goal representations, the points were connected in order of 1 (bigger and redder)– 2 (bigger and greener)– 4 (smaller and greener)– 3 (smaller and redder). Area and perimeter were then calculated using Python package Shaply.

The calculation of circularity index for sample stimuli and responses was similar to that of the goal space except for there were 9 conditions/points to consider (3 bins per stimulus feature). For the sake of constructing a close-formed geometric shape within the neural subspace for which circularity index could be calculated, the medium color value was not used, leading to 6 effective coordinates in the stimulus feature space.

**EEG: Time course of individual circularity index.**   To calculate the time-resolved individual circularity index, data was averaged within an 80-ms sliding window which were non-overlapping, resulting in 60 points per trial. Trial-wise data at each time point was first averaged within condition before PCA. The individual data matrix was then projected onto the first 2 PCs to acquire their coordinates in the 2D neural space and to calculate their circularity index. In order to increase the robustness of the estimation of circularity, trials were resampled in a stratified manner for 10 times and the above procedure was repeated. The resulting time series were averaged to derive the final individual circularity values.

**EEG: Cluster-based permutation test for circularity index time course.**   To test the significance of time-resolved circularity index values, trial labels were shuffled within participants before repeating the PCA and circularity calculation 5,000 times. Within each iteration, the trials were also resampled 10 times to remain consistent with the calculation of individual circularity. The individual shuffled time courses were averaged to generate a group-level null distribution. The true group circularity mean at each time point was compared to the resulting null distribution to derive $p$-values, with the cluster-forming threshold set at α = 0.05. Next, we repeated the previous step for the shuffled time courses, and each time the size of the largest cluster (defined as continuous supra-threshold time points) was taken to form the null distribution of cluster-level statistic, of which the 95th percentile was used as the cluster-level threshold.

**EEG: Behavioral relevance analysis based on trial splitting.**   For goal geometry, the bottom and top quartiles (i.e., 25%) of each participant's trial-wise data were selected based on the magnitude of combined response error to compute individual circularity indices corresponding to good and bad trials, respectively. The 2 sets of data were projected to the subspace identified by a PCA trained on the whole data set for a fair comparison of trials from both

quartiles. The difference in circularity indices between good and bad trials was then calculated. For stimulus-specific geometry, we instead used a median-split of the trials, as the quartile approach could not cover all unique conditions needed for the calculation. Statistical significance of the time course of individual circularity difference was assessed by cluster-based permutation in a similar fashion as described above, except that the null distribution was generated by randomly swapping the good and bad trial labels and calculating the circularity difference.

**EEG control analysis: Time-resolved representational similarity analysis (RSA).** To validate the goal representations in the 2D space revealed by PCA, we additionally performed an RSA analysis [69] on the individual level. For each participant, we tested whether a 2D and a conjunctive RSA models of task goals would explain the data. The 2D RSA model assumed that the pairwise representational distances between goals should resemble the 2D geometry as revealed by PCA. Specifically, the model representational dissimilarity matrix (RDM) for 2D goals (S2A Fig) was constructed by counting the number of same values across the 2 goal dimensions (i.e., the hamming distance: 0 for same goals, 0.5 for matching on only 1 dimension, and 1 for 2 goals with non-overlapping values). The conjunctive model assumed all goals are equidistant, resulting in a distance of 1 between- and of 0 within-condition. We included both task goals and stimulus features to define unique conditions in order to enrich the number of condition pairs, as with only the binary goal dimensions, number of valid pairs would be too low to allow stable estimates of model-neural correlations. For data RDM, cross-validated (4 folds) correlation distance was computed from the standardized data, averaged within an 80-ms non-overlapping sliding window. Comparison between the neural and model RDMs was performed using MNE-Python's function *mne_rsa.rsa*. Given the correlated nature of the 2 RSA models, we estimated the model-neural similarities both separately for each model (using Spearman's rank correlation as metric), as well as jointly in a competitive manner (using partial-Spearman as metric) to provide a comprehensive view of the results. Significance of RSA time courses from all individuals were tested against 0 using MNE-Python function *permutation_cluster_1samp_test*, except for the partial correlation result where significance was tested at each time point independently without cluster-based corrections.

**EEG control analysis: Motor-related neural geometry.** To ensure that the 2D goal geometry was not caused by signals related to motor preparation, a series of control analyses were conducted. The motor-specific conditions were defined by subtracting the initial values of the response object (which were randomly drawn for a uniform distribution) from the values of the final response. The signs of the resulting feature differences indicated the directions of adjustment the participants performed on that trial (e.g., to make the object bigger and redder from the initial status, one needed to press the buttons corresponding to "bigger" and "redder"), which were taken as the trial label specific to motor preparations. The following steps to calculate individual circularity index time course (for significance testing) were all identical to the procedures stated above. Chi-square tests for independence between possible motor preparation signals and task goals were performed using the Scipy library.

**EEG control analysis: Neural geometry of incorrect memory.** To examine whether the observed difference between good and bad trials was due to participants' incorrect memory of the goals (i.e., holding the wrong goal in mind on bad trials), we conducted a circularity analysis using only trials in which adjustment direction was incorrect. Specifically, we labeled the goal type based on participants' responses: if the direction of the response (redder, greener, bigger, or smaller) differed from the cued goal, we characterized these trials as wrong adjustment trials and used the actual direction of adjustment to relabel the condition. All subsequent steps followed the procedure in the main analysis for calculating the circularity index.

**EEG: Coherence analysis.** To understand how frontal control signals modulated goal-related representations, we calculated a coherence measure to investigate time-resolved connectivity between frontal and posterior channels. In particular, frontomedial theta oscillation (FMT) has been considered to be the medium through which executive control mechanisms orchestrate working memory contents [55,70]. To this end, coherence values were computed across all time points with the MNE-Python *spectral_connectivity_epochs* function, which involved calculating the cross-spectral densities, and estimating pairwise coherence in the range of 4 to 7 Hz between the seed frontal channel (Fz) and every posterior channel before being averaged. We chose the metric of weighted phase lag index (wPLI) as it is more robust against artifacts dues to volume conduction and noise [71]. Significant clusters in the baseline-corrected coherence values were determined using the MNE-Python function *permutation_-cluster_1samp_test* (one-tailed). To test whether significant FMT clusters during delay were functionally related to strength of representational geometries, the significant group clusters were taken as frequency and temporal masks to select coherence values from every individual's result, which were correlated with the circularity indices at each epoch of the trials, respectively. As a control analysis, we also repeated the above analysis on delta band (1 to 3 Hz).

**FMRI: General linear models (GLMs).** We fit a single-trial-level univariate GLM to extract neural response estimates for each delay period. For each functional run, the design matrix included task regressors representing separate periods of a trial and trial-wise regressors associated with the period of interest. For example, for modeling beta series of the first delay, the task regressors would consist of Goal cue (500 ms), Sample (600 ms), Delay 2 (8,400 ms), Response (5,000 ms), and trial-wise regressors for Delay 1 (i.e., Delay1_trial1[5500 ms], Delay1_trial2). The procedure was repeated for other time periods of interest. Thus, each trial was separated out into its own condition within the design matrix [72]. Additionally, the model included 6 head-motion regressors, 3 global signals from CSF, white matter and whole-brain, and 3 trend predictors from a polynomial drift model. Task regressors were convolved with the SPM canonical hemodynamic response function (HRF) and its time derivative. Functional data were standardized, high-pass filtered at 0.01 Hz and spatially smoothed with a 6-mm FWHM kernel. The resulting trial-wise beta series were brought forward to subsequent analyses.

To improve the separability of delay activity from stimulus- and probe-driven activity at the expense of sacrificing amplitude estimation [73], we performed an alternative GLM approach where we modeled the delay period as a single impulse response function. Specifically, a short regressor of 100 ms was placed at the middle of each delay period and convolved with HRF and its time derivative. All other steps remained the same as the previous whole-delay GLM. Of note, the impulse was only used to model the event for which trial-wise beta was extracted, while other irrelevant events were still modeled as boxcar to prioritize accurate magnitude estimation.

**FMRI: Circularity index with a searchlight procedure.** Consistent with the EEG analyses, for each participant, we computed individual circularity index by conducting PCAs on averaged single-trial beta coefficients within each unique condition (goals or stimulus bins). This was further combined with a searchlight procedure [74], allowing us to identify significant 2D task-related geometries across the whole brain. Using the *SearchLight class* in Nilearn, a spherical patch of 9-mm radius centered on each voxel was constructed within the whole-brain gray matter mask. Results from the whole-brain searchlight were visualized with SUMA in AFNI.

**FMRI: Cluster-based multiple comparison correction.** We carried out group-level cluster-based multiple comparison correction for the whole-brain circularity index. We first generated a null distribution of the group mean by running the above searchlight procedure with

shuffled trial labels 100 times for each participant. Then, the permuted maps were bootstrapped from each participant 10,000 times to create a null distribution of group-level mean [75]. Thirdly, the true group-level statistics (averaged across individual maps) were compared to the null distribution to compute the voxel-wise $p$-values. All voxels that passed the significant threshold ($\alpha = 0.01$) and formed clusters, defined as neighboring by face, were identified. Finally, we determined cluster-level threshold by iteratively drawing group-level shuffled maps from the null distribution to obtain the largest continuous cluster size using the same criteria in Step 3. This was repeated 10,000 times to generate a null distribution of cluster statistics, from which the 95th percentile value was used as the cluster threshold.

**FMRI: Test of cluster robustness using bootstrapping.** Moreover, to enhance the robustness of the searchlight results (which was ran once per participant due to the computational load), we additionally repeated the circularity analysis in each cluster, similar to the EEG analysis. Specifically, we adopted trial bootstrapping using all voxels within an ROI and repeatedly calculated the circularity for 20 times. The averaged cluster circularity indices were statistically assessed by comparing to a null distribution generated in a similar fashion as the true data. In the main text, only ROIs that have been revealed by the whole-brain searchlight and passed the test of robustness were shown and included in further examination. The original whole-brain searchlight results were shown in S8 Fig.

**FMRI: Beta series functional connectivity (FC).** Following the result of the searchlight analyses, we investigated interregional connectivity between the 3 largest significant clusters from the Delay 1 and 2 goal circularity and the Delay 2 stimulus circularity maps. Beta values estimated from the trial-wise GLM procedure described above were entered into a correlation estimation using the *ConnectivityMeasure* class in Nilearn. Specifically, parameter estimates were averaged within each cluster and correlation coefficients were calculated between every pairwise regions. The number of FC tested in subsequent correlation analyses varied based on our specific hypotheses: for correlation between FC and goal circularity, all connections between Delay 1 and Delay 2 goal-related clusters were included ($3 \times 3 = 9$ connections), since we specifically were interested whether FC is related to the relay of goal information. For stimulus and response circularity, connections between Delay 1 and 2 goal-related and stimulus- or response-related clusters were included ($6 \times 2 = 12$ connections each). All pairs included above were used in a correlation analysis with behavioral measure to assess the relationship between FC and memory performance. We also reported Bayes factor (BF10) in this section, as it provides an alternative perspective to evaluating the evidence for one hypothesis against another. BF10 was calculated using the python package Pingouin [76].

**FMRI control analysis: RSA.** Similar to the EEG RSA analysis, an RSA was conducted on the trial-wise delay-period beta estimates using a whole-brain searchlight procedure. Definition of the model RDMs was the same as used in the EEG analyses; for neural data RDM, correlation distance was computed from the standardized beta values using a customized leave-three-run-out cross-validation (due to a single run not covering all conditions). Comparison between the neural and model RDMs was performed using MNE-Python's function *mne_rsa.rsa* using both Spearman and partial Spearman rank correlation as metrics. Significance of correlations were tested in the same cluster-based multiple comparison correction procedure described above.

## Supporting information

**S1 Fig. Absolute error distributions for each participant.** (A) Individual absolute error distributions for color (left) and size (right) responses in EEG data. Each dot represents average absolute error from individual participant. Error bar represents 95% confidence interval. (B).

Same conventions as (A) but with results from fMRI data. Data and code that support these findings are available at: https://doi.org/10.57760/sciencedb.16868.
(TIF)

**S2 Fig. Representational similarity analysis results for EEG. (A)** Illustrations of representational distance matrices (RDMs) for 2D and conjunctive models. **(B, C)** Time-resolved cross-validated RSA correlations with individual data using the 2D and conjunctive goal models, estimated separately. Data was averaged temporally within a non-overlapping 80-ms sliding window. Red horizontal lines denote significant time points (α = 0.05) corrected using a cluster-based permutation test. Error bars represent 95% confidence interval. **(D, E)** Same as above but data-model similarities were estimated jointly in a competitive manner using partial Spearman's rank correlation. Red horizontal lines denote significant time points without cluster-based correction. Data and code that support these findings are available at: https://doi.org/10.57760/sciencedb.16868.
(TIF)

**S3 Fig. Circularity index for motor responses and incorrect goals in the EEG experiment.** (A) Individual circularity time courses of motor response signals for frontal (upper) and posterior (lower) channels. (B) Individual circularity time courses for frontal and posterior channels using incorrect trials and condition labels, defined as those in which participants adjusted the sample stimuli to a different direction from the task goals. Error bar denotes SEM. No significant time points were found using a cluster-based permutation test. Data and code that support these findings are available at: https://doi.org/10.57760/sciencedb.16868.
(TIF)

**S4 Fig. Whole-brain fMRI searchlight results using delay activity estimated with an impulse response function.** Significant ROIs showing 2D goal geometry in Delay 1 (yellow) and 2 (blue). Results were obtained by estimating delay activity using an impulse response function. Cluster-forming threshold was set to α = 0.01 and cluster-level threshold to α = 0.05 (same as the main result). Data and code that support these findings are available at: https://doi.org/10.57760/sciencedb.16868.
(TIF)

**S5 Fig. MNI y coordinates of fMRI searchlight cluster centers.** Distribution of MNI y coordinates (posterior to anterior axis) of all goal-specific clusters in the fMRI searchlight analysis. Each circle represents the center of a cluster and the diamond-shaped point represents the averaged value for each delay period. Data and code that support these findings are available at: https://doi.org/10.57760/sciencedb.16868.
(TIF)

**S6 Fig. RSA whole-brain searchlight results for fMRI. (A, B)** 2D and conjunctive representations, estimated separately for each model and subjected to cluster-based correction (cluster-forming threshold = 0.05 and cluster-level threshold = 0.05). A threshold of 50 voxels was applied to all statistical maps for visualization. **(C, D)** Same as above but data-model similarities were estimated jointly in a competitive manner. Data and code that support these findings are available at: https://doi.org/10.57760/sciencedb.16868.
(TIF)

**S7 Fig. Group-level 2D neural geometries in all searchlight ROIs in the fMRI experiment.** (**A**) Representational structures in the identified task goal subspace for Delay 1 activities. Each individual's condition-averaged data matrix was horizontally concatenated before applying PCA. Each colored dot represented a unique condition and were connected in the same order

as in the corresponding conceptual space. *C* denotes circularity index. **(B)** Same as **(A)** but for Delay 2. **(C)** Same as **(A)** but for sample stimulus geometry (using 6 conditions). **(D)** Same as **(C)** but for response stimulus geometry using participants' answers as feature values. Data and code that support these findings are available at: https://doi.org/10.57760/sciencedb.16868. (TIF)

**S8 Fig. Results for circularity robustness test for all searchlight-identified regions in the fMRI experiment.** Individual circularity was repetitively calculated and averaged using trial bootstrapping within each ROI. Orange and cyan color denotes ROIs that were identified by the circularity searchlight but did not pass significance threshold in robustness test in Delay 1 and 2, respectively; yellow and blue denotes those that were significant in both searchlight analysis and robustness test ($\alpha = 0.05$). Data and code that support these findings are available at: https://doi.org/10.57760/sciencedb.16868. (TIF)

**S1 Table. Correlation of delta band coherence with 2D geometries.** All *p*-values were FDR-corrected. (DOCX)

**S2 Table. Result summary of whole-brain searchlight for stimulus- and response-specific 2D geometry.** (DOCX)

**S3 Table. Statistics of pairwise correlation between functional connectivity and goal circularity (within Delay 2 ROI).** (DOCX)

**S4 Table. Uncorrected statistics for correlation of functional connectivity with response error.** (DOCX)

## Acknowledgments

We would like to thank Xuemei Zeng for help with data collection and Dongping Shi for discussion on the development of the circularity index.

## Author Contributions

**Conceptualization:** Mengya Zhang, Qing Yu.

**Formal analysis:** Mengya Zhang.

**Funding acquisition:** Qing Yu.

**Investigation:** Mengya Zhang.

**Methodology:** Mengya Zhang, Qing Yu.

**Supervision:** Qing Yu.

**Visualization:** Mengya Zhang.

**Writing – original draft:** Mengya Zhang, Qing Yu.

**Writing – review & editing:** Mengya Zhang, Qing Yu.

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
