## [Editor Report · Decision Letter 0]

6 Dec 2023

Dear Dr Yu, 

Thank you for submitting your manuscript entitled "Low-dimensional neural geometry underlies distributed goal representations in working memory" for consideration as a Research Article by PLOS Biology.

Your manuscript has now been evaluated by the PLOS Biology editorial staff as well as by an academic editor with relevant expertise and I am writing to let you know that we would like to send your submission out for external peer review.

Once your full submission is complete, your paper will undergo a series of checks in preparation for peer review. After your manuscript has passed the checks it will be sent out for review. To provide the metadata for your submission, please Login to Editorial Manager (https://www.editorialmanager.com/pbiology) within two working days, i.e. by Dec 08 2023 11:59PM.

Kind regards,

Christian

Christian Schnell, PhD

Senior Editor

PLOS Biology

cschnell@plos.org

---

## [Decision Letter · Decision Letter 1]

13 Mar 2024

Dear Dr Yu,

Thank you for your patience while your manuscript "Low-dimensional neural geometry underlies distributed goal representations in working memory" was peer-reviewed at PLOS Biology and apologies again for the long delay in sending our decision. It has now been evaluated by the PLOS Biology editors, an Academic Editor with relevant expertise, and by several independent reviewers. 

In light of the reviews, which you will find at the end of this email, we would like to invite you to revise the work to thoroughly address the reviewers' reports.

As you will see below, the reviewers are interested in your study but mention several major and serious concerns about the novelty of the study, regarding the methodological approach (composition of the data matrix, circularity in the design and methodology), and about the interpretations made from the work. Therefore, we are not yet sure whether your study is suitable for PLOS Biology and cannot make a decision about publication until we have seen the revised manuscript and your response to the reviewers' comments. Your revised manuscript is likely to be sent for further evaluation by all or a subset of the reviewers.

**IMPORTANT - SUBMITTING YOUR REVISION**

*Re-submission Checklist*

*Published Peer Review*

*PLOS Data Policy*

*Blot and Gel Data Policy*

Sincerely,

Christian

Christian Schnell, PhD

Senior Editor

PLOS Biology

cschnell@plos.org

REVIEWS:

Reviewer #1 (Runhao Lu): The manuscript describes low-dimensional neural geometry underlying abstract- (goal) and concrete- (stimulus) related representations in working memory. Using EEG and fMRI, the authors found converging evidence that the goal information first emerged in frontal areas for maintenance and then transferred to posterior areas for implementation, and the fidelity of the goal geometry is related to participants' performance. Overall, this is a neat study with lots of interesting and solid results and the manuscript on the whole is close to being publication-ready. I would happy to see the following issues answered/addressed.

1. I like the method of using the state-space analysis here, but this method seems originally just suitable for single-neuron studies, in which case the 'population level' is more meaningful as it refers to multiple neurons. While in EEG/fMRI situations, they refer to multiple electrodes/voxels, which may not have a direct neurophysiological meaning. Thus, it would be nice if the authors can describe a bit more about their thoughts of using this method under the situation of EEG/fMRI and perhaps also mention its potential limitations.

2. In the EEG state-space analysis, it seems a bit strange to particularly select the 3600-3800 ms time window to show the posterior geometry during late Delay 2, as it's neither pre-defined nor defined by the time-course results. Could the authors explain the reason about this choice?

3. It seems a bit abrupt to introduce the theta coherence as a candidate mediating the backward transfer, as all previous results are based on voltage rather than frequency power/phase. I acknowledge that authors did mention theta coherence in the Introduction, but the theoretical link between theta coherence and other results still seems not very strong. Also, some more interpretation on this point in Discussion would be nice.

4. Why the authors only focus on theta band (but not e.g. delta) as Figure 4A also showed significant effects in delta band? Also, do the authors think the backward transmission is specifically mediated by theta?

5. The authors chose the Prefrontal and Visual ROIs for the fMRI analysis, based on the EEG results. However, posterior EEG electrodes are like to reflect both parietal and occipital activities. Considering task-relevant information is decodable throughout the frontoparietal network, I was curious about whether parietal control regions, such as the intraparietal sulcus (IPS), also contribute to the goal information coding. If possible, having some results for an IPS ROIs might be an interesting bonus.

6. It is potentially very interesting that the authors found converging results for both theta phase-based connectivity and fMRI connectivity. Do the authors have any further results or discussion about their relationship or how similar they are?

Minor:

1. In the first paragraph, it would be nice to clarify/define what is 'abstract task information'.

2. Add references for the sentence starting from line 58: 'These distributed task representations are enhanced during goal implementation…'

3. It should be with cautious to use 'no evidence' (e.g. line 337) for the results with p > 0.05, given the authors were not using Bayes factors.

Reviewer #2: In this work, the authors examined the nature of neural signals during a working memory task. The task was fairly complicated with a 2x2 task rule involving two features: size and color. Subjects need to update a target stimulus with the task rule to make an adjustment response. Neural data (both EEG and fMRI) were analyzed for both the rule cue and the stimulus maintenance, focusing on low-dimensional representational geometry. The main findings are that low-dimensional signals first arise in frontal sites and then moved to posterior sites; frontal-posterior interaction (esp. via theta coherence) appears to mediate such transfer; and that these signals correlate with behavior. Overall these results show how abstract goals are represented in the brain and how they are communicated among brain regions and their behavioral relevance.

Overall I find this work quite novel and exciting. It has novel analyses and interesting results. There are also a lot of results, not the least of which were two independent experiments (EEG and fMRI). It is somewhat rare in my estimate to have such amount of data in a single paper. I commend the authors for this work. At the same time, I do have some concerns about the logic, the analysis method, and the interpretation, which I think the authors should address to solidify this work.

Major comments:

1. I find myself wanting to have a more intuitive explanation of what this main analysis looking for low-dimensional structure really means. Operationally, it is straightforward in that it seems to be looking for a neural replica of the factorial experimental design. Conceptually, does this mean that only neural signals that conform to such a geometry is meaningful? This approach seems to assume that the brain operates according to how an experiment is designed. Maybe this is a reasonable idea, but it also seems a bit circular: the researchers designed a low-dimensional factorial experiment, then they looked for neural signals that reflects such geometry, then concluded the low-dimensional signal underlies the behavior (e.g., the title of the paper). Are there alternative possibilities? For one thing, does this mean that there is no higher-dimensional information or is that not captured by this method? There is some text that seems to suggest the latter, a more explicit discussion of the logic and potential limitations would be helpful.

2. Related to the above, I'm particularly struck by the finding that posterior electrodes also showed a low-dimensional geometry (discussed in the second paragraph on p. 20). But earlier on, it was presented that sensory representations are thought to be high-dimensional, which seems to imply that different results would be found for frontal vs. posterior data. I can't quite wrap my head around this.

3. Method-wise, a somewhat odd choice of the analysis pertains to the way data were combined across subjects. It seems the data matrix are simply concatenated horizontally, which is analogous to running only a single subject with a lot more conditions, if I understand correctly. Is this correct? More explanation/citation of this method, and/or some simulation exercise, would help.

4. Regarding the analysis on frontomedial theta, Fig 4A shows that the majority of power in coherence is below 4Hz, so it seems odd to only use data from 4-7Hz, which is the nominal theta band, but it seems to miss the important stuff. Perhaps there is also frontomedial delta?

5. The fMRI GLM analysis modeled trial-wise individual event of interest, while the rest of the events were lumped together. This procedure was repeated for each event, to arrive at estimates for each event for each trial. I'm not too sure about this method, especially given the timing of events are fixed across trials. Rule number 1 in event-related fMRI is to temporally jitter events to allow separation of BOLD response in GLM. Are the authors confident that their fMRI data allow estimate of individual events on single trials?? 

6. The authors stated that the fMRI connectivity analysis showed "a robust bilateral positive relationship" between prefrontal and dorsal visual ROIs (line 481-483) and referred to Fig 6B. However, this is not obvious at all from inspecting Fig 6B.

7. This is a general comment. There are many results in this paper. However, there seem to be even more analyses that have been conducted. This is due to the richness of the data (e.g., different electrode groups, different events within a trial, different neural and behavioral measures). The authors reported statistically significant results, which are probably only a small subset of all possible tests. This does leave one wonder if some of these results (especially correlations) could be accidental. Some control/correction for multiple statistical tests are needed. However, I don't find any mention for such correction. In general, I think the authors should give some assurance of the robustness of their results.

Reviewer #3: Using EEG and fMRI, this study investigates the representation of task and task specific information in the human brain during a goal-directed visual working memory task. It shows that a low-dimensional neural geometry of goal information first appears in frontal cortex and then transferred to posterior sensory cortex through theta coherence. Goal-directed stimulus processing is an important component of human cognition and has been studied by an extensive amount of prior work. Overall, I am unsure about the novelty of this study and how it may contribute to this vast literature beyond what we already know. The review of the literature is also missing some key information. Some of the assumptions and manipulations in the study are made without detailed justifications. I also have some questions regarding data analyses.

(1) I am surprised that there is not an extensive review of the existing literature regarding neural synchrony during task performance between frontal and posterior regions, especially from monkey neurophysiology, such as Salazar et al. (2012, Science) and Siegel et al. (2015, Science). This existing literature already shows that goal information forms in frontal cortex and controls stimulus processing in posterior cortex. As such, I am not sure the main conclusion of the present study is novel. There is also an extensive literature regarding task representation in frontal cortex (e.g., the extensive work from Badre et al.). It is unclear how the present study add to that body of work.

(2) It is stated in the intro that "(y)et within working memory research, the mechanisms of how neural codes of abstract task information and specific stimulus contents collectively support goal-directed behavior remains an open question". What is special about working memory? Why can't we apply what we already know about goal-directed processing from the existing literature to working memory? Why should task information representation in working memory differ from other types of processing? I found the motivation of the study to be weak and does not point to a concrete research question beyond rather vague general statement.

(3) What is a low-dimensional neural subspace vs a high-dimensional one? How is it exactly defined? The task in the present study only involves a limited set of manipulations. So it is expected that the task structure would be low-dimension. Perhaps more explanations can be provided here. It is also presumed that task should form a two-dimensional structure in regions involved in task processing. What is the justification for this assumption? Task could be represented in an integrated manner in some brain regions rather than forming a two-dimensional structure, especially with learning. This possibility and its implications should be discussed.

(4) I don't fully understand the circularity index. Has it been used in other published studies to document orthogonal representations? Why is it a valid quantitative way to characterize the geometry? Why not just calculate the vector angles and see if it is close to 90 degrees? Why would a larger circularity index indicate higher similarity between the two structures? More detailed descriptions and proofs would be very helpful here. If the geometry is a rectangle rather than a square (i.e., with the two dimensions forming less orthogonal representations), what would the circularity value be? Also chance level circularity of .74 is rather higher, and only slightly lower than the significance circularity of .77 reported. Why is chance level circularity so high? Why not just use the RSA measures reported in lines 238-248?

(5) Could the behavioral correlation result reported in lines 250-262 simply due to participants forgot the goal in some trials and made the wrong adjustment? If so, this would not be a surprising result.

(6) Is it valid to bootstrap the data to achieve a more stable estimation for the difference in 2-D geometry strength (line 269-270)? The whole point of statistical test is to compare the amount of effect with respect to the amount of noise. If noise is artistically reduced by bootstrapping, then the procedure basically artificially boost the strength of the effect. 

(7) It is argued that in line 301 that the significant result of stimulus geometry obtained in Delay 1 was likely due to sporadic fluctuations. This is rather post hoc. How do we know other significant results are not due to such sporadic fluctuations?

(8) I don't fully understand why the goal representation shown cannot reflect motor related processing. It could still reflect abstract motor plan that is being formed before an actual motor response is executed.

(9) Group-concatenated data are subjected to the shortcomings of the fixed-effect analysis in that a few of the participants can drive the entire effect. Only results from random-effect analysis in which data are not concatenated should be reported.

(10) I am wondering how accurate EEG signal source localization is. Can we be sure that frontal sensors only have signals originate from frontal cortex?

(11) It took me a while to understand the task. Some more detailed descriptions and justifications of the task are needed. There are basically a fixed number of sizes and colors in a two-dimensional space. When participants are cued with stimulus X, they have to move one step in this stimulus space in the direction specified by the goal. This task requires long-term memory learning and retrieval of the stimulus space in addition to holding both task goals and stimuli in working memory. There was no motivation and justification for the task design. Why is this specific paradigm chosen?

---

## [Decision Letter · Decision Letter 2]

3 Sep 2024

Dear Dr Yu,

Thank you for your patience while we considered your revised manuscript "Low-dimensional neural geometry underlies distributed goal representations in working memory" for consideration as a Research Article at PLOS Biology. Your revised study has now been evaluated by the PLOS Biology editors, the Academic Editor and the original reviewers.

In light of the reviews, which you will find at the end of this email, we are pleased to offer you the opportunity to fully address the remaining points from the reviewers in a revision that we anticipate should not take you very long. We will then assess your revised manuscript and your response to the reviewers' comments with our Academic Editor, although might need to consult with the reviewers, depending on the nature of the revisions.

**IMPORTANT - SUBMITTING YOUR REVISION**

*Resubmission Checklist*

*Published Peer Review*

*PLOS Data Policy*

Sincerely,

Christian

Christian Schnell, PhD

Senior Editor

PLOS Biology

cschnell@plos.org

REVIEWS:

Reviewer #1 (Runhao Lu): The authors have niced answered my questions. I'm happy with the current version to be published.

Reviewer #2: The authors have addressed most of my comments. I appreciate the effort they made in the revision. I think the manuscript has improved significantly.

However, I do feel that my first comment in the original review was not completely addressed, regarding the circularity of the interpretation, i.e., quoting from the original review, "the researchers designed a low-dimensional factorial experiment, then they looked for neural signals that reflects such geometry, then concluded the low-dimensional signal underlies the behavior." In many parts of the paper (e.g., title, abstract, parts of the discussion), there is an emphasis on low-dimensional representation. If I understand it correctly, this approach will find high-dimensional signal if the task is designed to have a higher dimension. It seems more appropriate to talk about something like "task-congruent neural geometry", instead of "low-dimensional neural geometry". I think the central story won't change, i.e., task-related signals arise in frontal areas first and then propagate to posterior areas, which is interesting in itself. What I'm a little confused is about the absolute dimensionality of the representation (whether it's low or medium or high), which seems to be a secondary/orthogonal issue and cannot be pinned down by the current study. I think the authors need to de-emphasize the absolute dimensionality of the results, or further justify their rationale.

Lastly, in the spirit of open science and transparency, I would strongly encourage the authors to share their data and code on a public repository. Given the richness of the data, this would also benefit the field for others to explore this dataset for additional findings that might be interesting.

Reviewer #3: The authors have done a good job addressing the majority of my comments. However, I still have some remaining concerns (one major) that dampen my enthusiasm for the study.

Major:

In the previous round of review, I commented that:

"It is also presumed that task should form a two-dimensional structure in regions involved in task processing. What is the justification for this assumption? Task could be represented in an integrated manner in some brain regions rather than forming a two-dimensional structure, especially with learning."

In the reply letter, the authors stated that:

"To illustrate this, we performed an RSA analysis using a conjunctive model (which assumes that all goals are equidistant without forming a 2-D structure) and compared it with the 2-D model in the main text. Interestingly, we found that the conjunctive model was significant in the posterior channels during goal presentation and maintenance (Goal cue and Delay 1). By contrast, the 2-D model was significant during Delay 2 towards Response, consistent with our circularity analysis (Response Figure 1). These results suggest that sensory representation of goals is more likely to be in the conjunctive format, while the implementation of goals is more likely to be in the 2-D format, in line with our proposal that the 2-D geometry prompts goal transfer and implementation."

However, I don't see any discussion of this result in the revised manuscript. The design of the present study created a low-dimensional task space. Although some brain regions show such a low-dimensional structure, since a conjunctive task structure is also possible as described by the authors, examining only the low-dimensional task structure and drawing conclusions solely from such results may not be justified. It basically says that only the low-dimensional goal representation matters and others do not. What is the basis for this approach and assumption?

Minor:

(1) PFC may simply represent abstract information that needs to be retained during WM delay, whether such information specifies a goal or not. In other words, PFC may not be representing goal information per se, but any abstract info that is relevant in a WM task. In the present study, such abstract info happens to correspond to the goal of the second phase of the study. It would be helpful to discuss this.

(2) On lines 239-241 and lines 479-480, the authors stated that the fidelity of the goal geometry in PFC was predictive of individual behavioral memory performance. This implies a causal connection between the two, which has not been established. Both the goal geometry in PFC and behavioral memory performance may be influenced by the overall level of attention, fatigue, motivation, etc., rather than geometry determining performance.

---

## [Decision Letter · Decision Letter 3]

1 Nov 2024

Dear Dr Yu,

My name is Luke Smith - I am an editor at PLOS Biology and am writing on behalf of my colleague, Christian Schnell, who is the handling editor for your study, but who is off on vacation this week. I am writing to send you the most recent reviewer feedback and our decision for your manuscript "Task-congruent neural geometry underlies distributed goal representations in working memory". This revised version of your manuscript has been evaluated by the PLOS Biology editors, the Academic Editor and by two of the original reviewers whose comments are appended below.

Both of the reviewers and our Academic Editor are all satisfied by the revision and they have suggested that we accept the study. However, before we can editorially accept your manuscript for publication, we have a few last data and policy related requests that will need to be addressed in another short revision. We therefore ask that you revise your manuscript in response to the following points. 

EDITORIAL REQUESTS: 

1) TITLE: We would like to suggest a minor change to the title, which we think will make it more accessible to our broad readership. If you agree, and think that this still captures the main message of your study, we suggest you change the title to something like: 

Abstract goals are represented in working memory in a task-congruent neural geometry

We are happy for you to refine the title further.

2) ABSTRACT: Per journal policy, we ask that you edit your abstract to make it clear that this study was done in human subjects. 

3) ETHICS STATEMENT: Please update the ethics statement in your 'materials and methods' section to indicate whether the study was conducted according to the principles expressed in the Declaration of Helsinki.

4) DATA: I saw that your data availability statement says "All data files are available from the Science Data Bank database." - but I did not see an accession number where we could access this data (apologies if I missed it somewhere). Can you please provide us with an accession number so that we can confirm that your underlying data is in compliance with our data policies? For more details on the PLOS Data Policy, which requires that all data be made available without restriction, see the following: http://journals.plos.org/plosbiology/s/data-availability

5) CODE: Per journal policy, if you have generated any custom code during the course of this investigation, please make it available without restrictions. Please ensure that the code is sufficiently well documented and reusable, and that your Data Statement in the Editorial Manager submission system accurately describes where your code can be found.

We expect to receive your revised manuscript within two weeks. 

*Published Peer Review History*

*Press*

Sincerely,

Luke

Lucas Smith, PhD

Senior Editor

lsmith@plos.org

PLOS Biology 

-on behalf of-

Christian Schnell, PhD, 

Senior Editor

cschnell@plos.org

PLOS Biology

Reviewer remarks:

Reviewer #2: The auhtors have addressed my comments statisfactorily. I think this is a well done study, congrats!

Reviewer #3: The authors have satisfactorily addressed all my comments.

---

## [Editor Report · Decision Letter 4]

21 Nov 2024

Dear Qing,

Thank you for your patience while we considered your revised manuscript "Task-congruent neural geometry supports abstract goals in working memory" for publication as a Research Article at PLOS Biology.

We have now done our editorial checks and your manuscript is almost ready for acceptance. There are just the following few things that we would like you to still address:

* DATA POLICY:

Regardless of the method selected, please ensure that you provide the individual numerical values that underlie the summary data displayed in the following figure panels as they are essential for readers to assess your analysis and to reproduce it: 1FG, 5D, S1AB and S5.

* I realized that there is some supplementary methods text in the supplementary information. Can you please move this to the main manuscript file? We do not have restrictions in terms of word count or number of references, and want to make it as easy as possible for readers to find the information they need. Please also add the references from this section to the main reference list if they are not already included.

* Regarding the title, I think your suggestion is alright, but I am still a bit concerned that many readers will be lost right at the beginning because they are unfamiliar with task-congruent neural geometry. Would it work for you to write it the other way around: "The representation of abstract goals in working memory is supported by task-congruent neural geometry"? Let me know what you think and please get in touch via email if you would to discuss further or have an alternative suggestion.

We expect to receive your revised manuscript within two weeks. 

*Published Peer Review History*

*Press*

Sincerely,

Christian

Christian Schnell, PhD

Senior Editor

cschnell@plos.org

PLOS Biology

---

## [Editor Report · Decision Letter 5]

29 Nov 2024

Dear Qing,

Thank you for the submission of your revised Research Article "The representation of abstract goals in working memory is supported by task-congruent neural geometry" for publication in PLOS Biology. On behalf of my colleagues and the Academic Editor, Frank Tong, I am pleased to say that we can in principle accept your manuscript for publication, provided you address any remaining formatting and reporting issues. These will be detailed in an email you should receive within 2-3 business days from our colleagues in the journal operations team; no action is required from you until then. Please note that we will not be able to formally accept your manuscript and schedule it for publication until you have completed any requested changes.

PRESS

Sincerely, 

Christian

Christian Schnell, PhD

Senior Editor

PLOS Biology

cschnell@plos.org